

# Size-resolved mixing state of black carbon in the Canadian high Arctic and implications for simulated direct radiative effect

John K. Kodros[1], Sarah Hanna[2], Allan Bertram[2], W. Richard Leaitch[3], Hannes Schulz[4], Andreas Herber[4], Marco Zanatta[4], Julia Burkart[5], Megan Willis[5], Jonathan P. D. Abbatt[5], Jeffrey R. Pierce[1],

[1]Department of Atmospheric Science, Colorado State University, Fort Collins, Colorado, 80523, USA
[2]Department of Chemistry, University of British Columbia, Vancouver, BC, Canada
[3]Climate Chemistry Measurements and Research, Climate Research Division, Atmospheric Science and Technology Directorate, Environment and Climate Change Canada, 4905 Dufferin Street, Toronto, ON, M3H 5T4, Canada
[4]Alfred Wegener Institute Helmholtz Center for Polar and Marine Research Bremerhaven, Bremerhaven, Germany
[5]University of Toronto, Department of Chemistry, Toronto, Ontario, Canada

*Correspondence to*: John K. Kodros (jkodros@atmos.colostate.edu)

**Abstract.** Transport of anthropogenic aerosol into the Arctic in the spring months has the potential to affect regional climate; however, modeling estimates of the aerosol direct radiative effect (DRE) are sensitive to uncertainties in the mixing state of black carbon (BC). A common approach in previous modeling studies is to assume an entirely external mixture (all primarily scattering species are in separate particles from BC) or internal mixture (all primarily scattering species are mixed in the same particles as BC). To provide constraints on the size-resolved mixing state of BC, we use airborne Single Particle Soot Photometer (SP2) and Ultra-High Sensitivity Aerosol Spectrometer (UHSAS) measurements from the Alfred Wegener Institute (AWI) POLAR6 flights from the NETCARE/PAMARCMIP2015 campaign to estimate coating thickness as a function of refractory BC (rBC) core diameter as well as the fraction of particles containing rBC in the springtime Canadian high Arctic. For rBC core diameters in the range of 140 to 220 nm, we find average coating thicknesses of approximately 45 to 40 nm, respectively, resulting in ratios of total particle diameter to rBC core diameters ranging from 1.6 to 1.4. For total particle diameters ranging from 175 to 730 nm, rBC-containing particle number fractions range from 16 to 3%, respectively. We combine the observed mixing-state constraints with simulated size-resolved aerosol mass and number distributions from GEOS-Chem-TOMAS to estimate the DRE with observed bounds on mixing state as opposed to assuming an entirely external or internal mixture. We find that the pan-Arctic average springtime DRE ranges from -1.65 W m$^{-2}$ to -1.34 W m$^{-2}$ when assuming entirely externally or internally mixed BC. Using the observed mixing-state constraints, we find the DRE is 0.05 W m$^{-2}$ and 0.19 W m$^{-2}$ less negative than the external mixing-state assumption when constraining by coating thickness of the mixed particles and by BC-containing particle number fraction, respectively. The difference between these methods is due to an underestimation of BC mass fraction in the springtime Arctic in GEOS-Chem-TOMAS compared to POLAR6 observations. Measurements of mixing state provide important constraints for model estimates of DRE.



# 1 Introduction

Over the last several decades, the Arctic has warmed at near twice the rate of the global mean (IPCC 2013). In addition to $CO_2$, short-lived climate forcers (such as black carbon and methane) may contribute to this increased rate of warming (AMAP, 2015). Black carbon (BC), in particular, may be an efficient warming agent in the Arctic as its potential to

decrease planetary albedo is magnified when BC is above a white surface, such as snow (e.g. Bond et al., 2013). BC can affect climate through aerosol-radiation interactions (e.g. the direct radiative effect, semi-direct radiative effect) or through aerosol-cloud interactions (e.g. the cloud-albedo indirect effect) (Boucher et al., 2013). The magnitude to which BC warms an atmospheric column through aerosol-radiation interactions is strongly sensitive to atmospheric concentration, altitude, and chemical mixing state (Bond et al., 2013). In this work, we focus on the mixing state of BC.

The population (or chemical) mixing state of BC refers to the degree in which BC particles are mixed with other aerosol species. In global and regional models where population mixing state is not explicitly tracked, BC is commonly represented as completely externally mixed (separate from other aerosol species) or a completely internally mixed (mixed together with other aerosol species). For BC mass in a given particle size range, the external mixture assumes the BC mass is divided up into particles of the same size as the non-BC particles. Conversely, in the internal mixture, the same BC mass is

divided up into a relatively larger number of particles with smaller BC diameters such that the total mixed particle has the representative diameter of the size range. In a full internal mixture, BC mass is spread among all particles. For a given BC mass, the internal mixing-state assumption will produce more absorption as the BC mass is spread amongst more particles giving rise to greater surface area and thicker coatings (Bond and Bergstrom, 2006; Seinfeld and Pandis, 2012). However, in the atmosphere, BC is often not mixed entirely as an internal or external mixture. For instance, Massoli et al. (2015) found

that 35% of the total non-refractory submicrometer mass is associated with BC-containing particles in coastal California during the CALNEX campaign, and recent studies of the Arctic aerosol suggest a smaller percentage of the aerosol particles contain BC (e.g. Raatikainen et al., 2015; Sharma et al., 2017). Thus, the assumption in aerosol models of a full internal or external mixture may lead to over- or under-estimates of BC absorption.

A second component of BC mixing state is the morphological mixing state. The morphological mixing state refers

to the distribution of chemical species within a particle, as well as the shape and location of BC within the aerosol particle. A common assumption for the morphological mixing state of BC is that the BC mass forms the core of a particle and the hydrophilic aerosol mass forms a shell around the particle (Bond et al., 2006; Lack and Cappa, 2010). The scattering component of the shell acts as a lens to focus more photons onto the core, thus increasing absorption (Borthen and Huffman, 1983; Bond et al., 2006). However, the degree of absorption enhancement is a strong function of the core diameter and shell

thickness. Theoretical calculations (known as "core-shell Mie theory") and laboratory studies have estimated enhancements in absorption for a given BC mass of a factor of 1.3 to 2 (Schnaiter et al., 2003; Schnaiter et al., 2005; Zhang et al., 2008). Based on these findings, Bond et al. (2006) recommends scaling BC absorption by a factor of 1.5 to account for the "lensing" effect in models that do not assume internal aerosol mixtures. However, field campaigns have found a wider



variation of absorption enhancements. While finding that 20-30% of BC particles had acquired a thick coating, McMeeking et al. (2011) and Subramanian et al. (2010) did not find a dependence of mass absorption cross section with coating. Similarly, Cappa et al. (2012) found an absorption enhancement of only 6%, and speculated that the BC may not be at the exact center of the particle. Conversely, Wang et al. (2014) found an absorption enhancement of 1.8 in China, similar to

laboratory findings. In addition, Sharma et al. (2017) found values of mass absorption cross section to increase with coating thickness with a steeper slope than theoretical calculations. Thus, accurate model estimates of BC absorption must rely on an understanding of sources of BC as well as all other aerosol components.

Aerosol number and mass concentrations in the Arctic have a strong seasonal cycle, with contributions from anthropogenic sources leading to a peak in accumulation-mode aerosol mass in the winter and spring (e.g. Quinn et al., 2007;

Croft et al., 2016). Efficient wet removal in the summer results in conditions that favor new particle formation and nucleation-mode aerosol (e.g. Garrett et al., 2011; Browse et al., 2012; Leaitch et al., 2013; Tunved et al., 2013; Croft et al., 2016a; Croft et al., 2016b; Willis et al., 2016). As there is little or no solar irradiance in the Arctic winter, it is essential to accurately simulate aerosol optical properties in the spring. Sources of BC in the Arctic include gas flaring and biomass burning; however, much of the atmospheric BC concentration is transported from lower latitudes (Xu et al., 2017; Schulz et

al., in prep). Using a combination of observations and a chemical transport model, Xu et al. (2017) found that in the Arctic spring much of the BC concentrations at higher elevations arrived from South Asia, while at lower elevations BC concentrations were transported from Asia and Eastern Europe. Sources of non-BC particles to the Arctic include direct marine sources, new particle formation from natural or possibly anthropogenic precursors, and transport from lower latitudes (e.g. Croft et al., 2016b; Wentworth et al., 2016; Willis et al., 2016).

Despite the dependence of aerosol absorption on BC mixing state, measurements of mixing state in the Arctic are limited. The Single Particle Soot Photometer (SP2) provides direct measurements of size-resolved refractory BC (rBC) mass for particle diameters in the range of approximately 75-700 nm. In addition, by the leading-edge fit method (Gao et al., 2007), the SP2 can provide estimates of coating thickness on rBC particles. Combining the size-resolved coated rBC measurements with a total aerosol size distribution provides information on the fraction of total aerosol number containing

rBC, thus providing a constraint on the population mixing state of rBC. Raatikainen et al. (2015) measured rBC mixing state properties with an SP2 in the Finnish Arctic winter, finding 24% of particles contain rBC with an average total particle to refractory rBC core diameter of 2. Similarly, Liu et al. (2015) found an average total particle to rBC core diameter of 2.25 (1.7 - 2.8) during aircraft campaign in the European Arctic in March. For biomass burning plumes, Kondo et al. (2011) measured a ratio of 1.3-1.6 during aircraft flights in the spring and summer in the Canadian Arctic. Finally, Sharma et al.

(2017) found a ratio of 1.4-1.25 at the Alert ground station in spring of 2012.

In this study, we present measurements of BC mixing state aboard the Alfred Wegener Institute (AWI) POLAR6 flights, as part of the Network on Climate and Aerosols: Addressing Key Uncertainties in Remote Canadian Environments (NETCARE) and Polar Airborne Measurements and Arctic Regional Climate Model Simulation Project (PAMARCMiP). We use these measurements as constraints on simulated aerosol mass and number concentrations to estimate the direct



radiative effect (DRE) in the springtime Arctic, and compare these estimates to the DRE calculated using bounding cases of completely external or internal mixing state-assumptions. While this study focuses on the DRE, we note that other aerosol-radiation interaction processes, such as the semi-direct effect, may also depend on BC mixing state. In Sections 2.1-2.3, we present the POLAR6 flight paths and size-resolved aerosol measurement setup. In Section 2.4, we discuss the chemical-

transport model and assumptions in the calculation of the DRE. In Section 3, we present observations of BC mixing state and implications for simulated DRE. We share our conclusions and study limitations in Section 4.

## 2 Methods

### 2.1 Flight overview and sample locations

As part of the Network on Climate and Aerosols: Addressing Key Uncertainties in Remote Canadian Environments

project (NETCARE, http://www.netcare-project.ca), and in collaboration with the Polar Airborne Measurements and Arctic Regional Climate Model Simulation Project (PAMARCMiP (Herber et al., 2012)), measurements of aerosol, trace gases and meteorological parameters were made in High Arctic spring aboard the Alfred Wegener Institute (AWI) Polar 6 aircraft, a DC-3 aircraft converted to a Basler BT-67 (Herber et al., 2008). Measurements on a total of 10 flights took place from 4 – 22 April 2015 based at four stations: Longyearbyen, Svalbard (78.2°N,15.6°E); Alert, Nunavut, Canada (82.5°N, 62.3°W);

Eureka, Nunavut, Canada (80.0°N, 85.9°W); and Inuvik, Northwest Territories, Canada (68.4°N, 133.7°W). Polar 6 allowed measurements from 60 – 6000 m, with a survey speed of ~75 m s$^{-1}$ (~270 km h$^{-1}$) with ascent and descent rates of ~150 m min$^{-1}$. This study uses observations from flights conducted at Alert and Eureka, between 7 – 13 April 2015. Table 1 presents the date, departure location, and total flight time for the six flights used here. Flight tracks are shown in the map in Figure 1.

Aerosol was sampled near-isokinetically through a stainless steel shrouded diffuser inlet, located ahead of the

engines. The inlet provided near unity transmission of particles 20 nm to ~1 µm in diameter at typical survey airspeeds, corresponding to a total flow rate of ~55 L min$^{-1}$. Bypass lines off the main inlet, at angles less than 90°, carried aerosol to various instruments described in Section 2.2 and 2.3. Aerosol was not actively dried prior to sampling; however, the temperature in the inlet line within the aircraft cabin was >15 °C warmer than the ambient temperature so that the relative humidity decreased significantly.

Periods when the aircraft was in-cloud were determined using data from a Forward Scattering Spectrometer Cloud Probe (FSSP-100, Droplet Measurement Technologies, Inc). The FSSP is an optical particle counter that detects droplets in the range of 2-50 µm in diameter. Any periods where a signal above noise was registered by the FSSP were determined to be in-cloud times and were removed from the dataset. In total the aircraft was in-cloud for ~0.3% of the flight time.



## 2.2 Measurement of refractory black carbon

### 2.2.1 rBC number vs rBC core diameter

Measurements of refractory black carbon (following the definition in Petzold et al., 2013) were made with a single particle soot photometer (SP2, Droplet Measurement Technologies Inc). The SP2 detects individual particles using an intra-cavity Nd:YAG laser operating at 1064 nm. As particles pass through the laser beam, those that contain a strongly absorbing component at 1064 nm (such as black carbon) are heated to incandescence. Light emitted by the incandescing fraction of the particle is detected by a pair of photomultiplier tubes. The peak amplitude of the thermal radiation emitted by the incandescing particle is proportional to the mass of refractory material in the particle (Moteki and Kondo, 2007; Slowik et al., 2007). In this work, size-selected Aquadag particles (Acheson Industries) were used as an external standard for mass calibration of the SP2. Measured Aquadag mobility diameters were converted to rBC (refractory black carbon) mass using the size-dependent effective densities reported by Gysel et al. (2011). Recent studies have shown that the SP2 is more sensitive to Aquadag than it is to ambient rBC (Laborde et al., 2012a; Moteki and Kondo, 2010). In order to account for this, we have scaled the slopes of the Aquadag-derived calibration curves by a factor of $0.70 \pm 0.05$. This scaling factor was derived from Figure 5 in Laborde et al. (2012a) which shows the response of the SP2 to Aquadag as well as the response of the SP2 to rBC from diesel exhaust, wood smoke, and ambient particles.

Two single particles soot photometers (referred to as SP2#1 and SP2#2) were used in this study. SP2#1 had a detection range of 0.40 - 323 fg rBC (equivalent to spherical diameters ranging from 75 to 700 nm at an rBC density of 1.8 g cm$^{-3}$ (Bond and Bergstrom, 2006)), while SP2#2 had a detection range of 0.40 - 9.37 fg rBC (equivalent to spherical diameters ranging from 75 to 220 nm at an rBC density of 1.8 g cm$^{-3}$ (Bond and Bergstrom, 2006)). SP2# 1 was used to measure rBC number, since it had a wider size detection range than SP2#2. SP2#2 was used to determine coating thicknesses as a function of rBC core diameter due to a misalignment of an optical detector needed for coating analysis in SP2#1.

### 2.2.2 rBC coating thickness vs rBC core diameter

As particles pass through the laser beam in the SP2 they elastically scatter light at 1064 nm. For particles that contain rBC, the particle may be heated and begin to vaporize before the scattering intensity reaches the peak value that an unperturbed particle would have reached. In this case, the unperturbed scattering amplitude can be retrieved by fitting the leading edge of the particle's scattering signal as described by Gao et al. (2007). In the leading edge fit method, the center position and width of a Gaussian (which reflects the laser beam profile) are fixed from the scattering profiles of non-incandescing particles collected during the preceding and following hour. The unperturbed peak scattering amplitude for an incandescing particle is then determined by fitting the measured scattering profile up to 5% of the peak scattering intensity using the fixed width and center position and allowing only the amplitude to vary. Fitting the profile only up to 5% of peak elastic scattering intensity allows the unperturbed peak amplitude to be retrieved. For an individual particle the uncertainty in



reproducing the unperturbed amplitude is ~70% (~20% uncertainty in optical size) but since this uncertainty is random it does not systematically impact our results.

     With the scattering amplitude determined by the leading-edge technique, and the measured rBC core diameter, a core-shell Mie model can be used to determine the optical diameter of the rBC-containing particles. In the Mie model we
used a refractive index of 2.26–1.26i for rBC (Moteki et al., 2010) and a refractive index of 1.5-0.0i for the coating species. The value of 1.5-0.0i is appropriate for dry sulfate and sodium chloride (Schwarz et al., 2008a, 2008b).

     Although the SP2 can measure individual particle rBC mass down to 0.40 fg (~75 nm VED) the elastic scattering optical detectors can only measure scattering from bare rBC particles with volume-equivalent diameters of ~120 nm or greater. As a result, when rBC cores have a diameter smaller than 120 nm only those cores with significant coatings will
produce a measurable elastic scattering signal. Additionally, in practice fewer than 90% of particles with a rBC core diameter of less than 140 nm are successfully assigned a coating. This is because the leading edge fit method uses two detectors to accurately determine the coating thickness. One of these is a two element scattering detector used to determine the position of a particle in the laser beam. With this two element detector, the scattering signal from a particle passing through the laser has a clear notch when it passes the gap between detecting elements and this notch position is used to locate
the particle in the beam. In some cases, rBC containing particles may evaporate prior to crossing the notch position and, and as a result, their coating thickness cannot be determined. This happens more frequently for smaller particles. As a result of the two caveats discussed above, there is a bias toward thicker coatings for rBC cores < 140 nm. To account for this bias, we calculate minimum, median, and maximum bounding cases for coating thickness for rBC core diameters less than 140 nm. A detailed discussion of the estimation of each case and the resulting impact on the DRE is included in the Supplemental
Information (SI). As we do not see a substantial difference in the DRE across the three bounding cases, we focus on the median value for the remainder of this text. The median estimate of coating thickness for rBC core diameters less than 140 nm is taken as the overall median coating thickness across core diameters 140 to 220 nm.

### 2.3 Measurement of total aerosol size distributions

     Total aerosol size distributions for particles with diameters in the range of 85 - 1000 nm were measured by an Ultra
High Sensitivity Aerosol Spectrometer (UHSAS, Droplet Measurement Technology Inc). The UHSAS is a laser-based aerosol spectrometer in which particles intercept the beam of a Solid-state Nd3+:Y LiF4 laser operating at ~1054 nm. Two sets of Mangin mirrors focus light scattered by the particles onto two detectors; one a high gain avalanche photodiode for detecting particles smaller than 250 nm and the other a low-gain PIN photodiode for detection particles larger than 250 nm. Counting efficiency for the UHSAS is >95% for particle concentrations < 3000 cm$^{-3}$. Further details of the instrument
operating principles can be found in Cai et al (2008). The UHSAS was calibrated in during operations using polystyrene latex spheres (PSLs). Details of the calibration and comparison with other in-flight measurements can be found in Leaitch et al. (2016) as well as Willis et al. (2016) and Burkart et al. (2017). One potential issue with using the UHSAS aboard an



aircraft is that, as the pressure changes during ascent and descent, the sample flow at the inlet of the chamber can deviate from the measured and regulated flow at the outlet of the chamber (Brock et al., 2011; Kupc et al., 2018) which can results in inaccurate particle concentration measurements. Brock et al. saw particle number deviations from a reference counter of ~10-15% on ascent and descent. In this study, comparison to counts of non-incandescent particles from the SP2, which was

on the same inlet line, suggest that particle counts deviated by <5%.

**2.4 Determination of the fraction of total aerosols containing rBC vs aerosol diameter**

To determine the fraction of total aerosol particles containing rBC as a function of size, we first determined the size distribution of the rBC-containing particles, this time accounting for both the rBC core diameter and the thickness of any coating material. This was done by taking the coating thicknesses as a function of rBC core size for each flight and applying

those coating thicknesses to the measured rBC core sizes from SP2#1. Once the coating thickness were applied to the rBC cores measured by SP2#1, the particles were then binned according to their total size (core diameter plus 2 times the coating thickness) to give the number distribution for rBC containing particles. The bin increments were set to match the increments from the UHSAS and the fraction of rBC containing particles was calculated for each bin by dividing the number of rBC-containing particles by the number of total aerosols. As with the coating analysis, this process was carried out separately for

each flight and the average of all flights were combined with GEOS-Chem-TOMAS simulations.

**2.5 Model overview**

**2.5.1 GEOS-Chem-TOMAS**

To simulate aerosol concentrations in the Arctic, we use the Goddard Earth Observing System chemical-transport

model, GEOS-Chem, version 10.01. We simulate April 2015 with 2 months of spin-up not included in the analysis. Transport in GEOS-Chem is driven by MERRA re-analysis meteorology fields. This version of GEOS-Chem uses a horizontal resolution of 4 degrees latitude by 5 degrees longitude with 47 vertical layers.

Aerosol microphysics is simulated using TwO Moment Aerosol Sectional (TOMAS) microphysics scheme (Adams and Seinfeld, 2002) coupled with GEOS-Chem (known as "GEOS-Chem-TOMAS"). The version of TOMAS used in this

study includes 40 size bins ranging from diameters of approximately 1 nm to 10 μm (the exact diameter depends on the density of the aerosol). TOMAS includes tracers for sulfate, BC, organic aerosol, sea salt, dust, and aerosol water. In this work, we will use the term "BC" when referring to simulated mass concentration and "rBC" when referring to measurements made with the SP2. Description of aerosol microphysics in TOMAS has been included in Adams and Seinfeld (2002), Lee et al. (2013), and Lee and Adams (2012). Croft et al. (2016) compares aerosol size distributions simulated by GEOS-Chem-

TOMAS to observations at Alert, Nunavut and Mt. Zeppelin, Svalbard and includes a description of the settings used here.

Global anthropogenic emissions are derived from The Emissions Database for Global Atmospheric Research (EDGAR) Hemispheric Transport of Air Pollution (HTAP) version 2.2 (Janssens-Maenhout et al., 2015). Following the





recommendation in Xu et al. (2017), we include BC and organic carbon emissions from gas flaring derived from the Evaluating the Climate and Air Quality Impacts of short Lived Pollutants (ECLIPSE) emission inventory (Klimont et al., 2017). Biomass burning emissions are from the The Fire Inventory from NCAR (FINN) for the year 2015 (Wiedinmyer et al., 2011). Dust aerosol emissions follow the DEAD scheme (Zender et al., 2003), while sea-salt aerosol emissions are based

on the scheme of Jaeglé et al. (2011).

**2.5.2 Direct radiative effect calculation**

The all-sky direct radiative effect is estimated using an offline version of the Rapid Radiative Transfer Model for GCMs (RRTMG; Iacono et al., 2008), following the online version implemented in GEOS-Chem (Heald et al., 2014). RRTMG treats clouds using the Monte Carlo independent column approximation (McICA; Pincus et al., 2003). Aerosol

optical properties are calculated using monthly averaged aerosol mass and number concentrations with refractive indices from the Global Aerosol Dataset (GADS). We use Mie code published in Bohren and Huffman (1983) to calculate aerosol optical depth, single scattering albedo, and asymmetry parameter. Available code includes Mie calculations for 2 concentric spheres (for use in core-shell morphologies). The use of monthly mean aerosol and cloud properties is a limitation of this study; however, we feel this is sufficient to explore the impacts of different BC mixing-state assumptions on the DRE.

**2.5.3 Description of aerosol mixing states**

We calculate the DRE with 5 BC mixing states, outlined in Table 2. Figure 2 is a schematic of the mixing states. We discuss a numerical example at the end of this section that follows Figure 2. In the "*external*" mixing state, all BC exists as a separate particle from the other aerosol species (Figure 2a). The "*external*1.5*" mixing state multiplies the BC absorption in the "*external*" mixing state by a factor of 1.5 to simulate absorption when BC is mixed internally, following the

recommendation in Bond et al. (2006b). As the "*external*1.5*" mixing state only modifies the absorption from the "*external*" mixing state, we do not explicitly animate it in Figure 2. Conversely to the "*external*" mixing state, the "*allCoreShell*" assumption treats all BC mass in each size bin as the core with other hydrophilic species forming a shell around the BC core (Figure 2d). The *external* and *allCoreShell* mixing states are bounding cases, assuming that 0 and 100%, respectively, of BC particles are mixed with the other aerosol components. The *external*1.5* mixing state is included as a reference point in

between the two bounding cases.

We compare these assumed mixing states to two mixing states based on the measurements described in Sections 2.2-2.3. The "$r_{shell}$-*constrained*" mixing state uses measurements from the SP2 of shell thickness as a function of core diameter to constrain BC mixing state (Figure 2b). Across the SP2 size range, we take the total BC mass in a given size bin simulated by TOMAS, and form BC cores with diameters given by the SP2 measurements. We then take the total scattering

mass simulated by TOMAS in the same size bin and form concentric surrounding shells based on the thickness measured by the SP2. Any remaining scattering aerosol mass forms a separate particle with no BC included. Outside of the SP2 size range, we retain the ratio of rBC core diameter to total particle diameter at the lower/upper edge of the SP2 range and apply




that to estimate core and shell diameters for the remaining TOMAS size bins. Using the measured rBC core diameter and shell thickness from the SP2, the resulting fraction of BC-containing particles for a given size range can be described by Equation 1:

$$fBC = \cfrac{\left(\cfrac{M_{BC}}{M_{tot}}\right)}{\left(\cfrac{\rho_{BC}\frac{4}{3}\pi\, r_{core}^3}{\rho_{shell}\frac{4}{3}\pi\, r_{shell}^3 + \rho_{BC}\frac{4}{3}\pi\, r_{core}^3}\right)} \tag{1}$$

Where $\rho_{shell}$ is the density of the shell, $r_{shell}$ is the shell thickness, $\rho_{BC}$ is the density of the BC core, $r_{core}$ is the radius of the core, $M_{tot}$ is the total aerosol mass, and $M_{BC}$ is the mass of BC. The numerator in Equation 1 describes the BC mass fraction of the aerosol population for a given size range, while the denominator describes the BC mass fraction for a single aerosol particle. The resulting fraction of BC-containing particles is thus a function of the BC mass fraction as well as the ratio of the

BC radius to the total particle radius. In our approach, values for $r_{shell}$ and $r_{core}$ are taken from the SP2 measurements while $M_{BC}$ and $M_{tot}$ are simulated by GEOS-Chem-TOMAS. The $r_{shell}$-constrained BC mixing state will not reproduce the measured fraction of rBC-containing particles unless the ratio of BC and scattering mass is similar to observed with the SP2 and UHSAS.

The second measurement-constrained mixing state, "fBC-constrained", uses the size-dependent fraction of all

particles containing rBC as the measurement constraint (Figure 2c). In a given size range, we separate total aerosol mass and number (simulated by TOMAS) into a population containing BC and a population not containing BC based on the measured fraction of particles containing BC. We then form BC cores and scattering shells based on the simulated mass-to-number ratio and assumed density for each species. Using the fraction of rBC containing particles measured with the SP2 and UHSAS, the resulting BC core diameter and shell thickness can be calculated by Equation 2 and 3:

$$d_{core} = \left[\frac{M_{BC}}{fBC \times N \times \rho_{BC} \times \frac{\pi}{6}}\right]^{1/3} \tag{2}$$

$$r_{shell} = \frac{d_{tot} - d_{core}}{2} \tag{3}$$

Where $d_{core}$ is the BC core diameter, $fBC$ is the fraction of BC-containing particles measured with the SP2 and UHSAS, $N$ is

the total particle number, and $d_{tot}$ is the total particle diameter in the given TOMAS size bin. Similarly, this method will not produce the measured rBC core diameter and shell thickness unless TOMAS simulates a similar ratio of BC to scattering mass as is observed. For both $r_{shell}$-constrained and fBC-constrained, we use the median measured values across all flights (see Supplemental Information for explanation), and we compare results of the minimum, median, and maximum cases




based on the uncertainties in SP2 measurements in the 100-140 nm diameter size range range in the Supplemental Information.

The different mixing states are depicted schematically in Figure 2 with the bold text highlighting the parameter being constrained in each case. As an example, we depict the TOMAS size bin for 250 nm diameter particles where GEOS-

Chem-TOMAS simulates a particle number concentration of 100 cm⁻³ and a ratio of BC to total-aerosol mass of 4%. In the *external* mixing-state assumption (Figure 2a), the 0.06 µg m⁻³ of BC mass is used to form pure BC particles with diameters of 250 nm (by definition, a shell thickness of 0 nm). This results in 4% of the particle number concentration being pure BC particles, while the remaining particles are composed of non-BC species also with diameters of 250 nm (for convenience, 1 out of 10 particles in the schematic depiction of *external* is pure BC). In the $r_{shell}$-*constrained* mixing state (Figure 2b), we use

the observed rBC core diameter and corresponding shell thickness from the SP2 measurements (in this example a core of 150 nm and a shell thickness of 50 nm) to allocate the same 0.06 µg m⁻³ BC mass into mixed particles with total diameter 250 nm. As the BC diameter is now only 150 nm, the resulting BC containing particle number fraction is 20%. In the *fBC-constrained* mixing state (Figure 2c), we instead constrain by the fraction of rBC-containing particles measured with the SP2 and UHSAS (30% in this example). To allocate the same BC mass onto 30% of particles, the BC core diameter has to equal

128 nm resulting in a shell thickness of 61 nm (to create a particle with total diameter of 250 nm). Finally, to allocate the same BC mass onto all particles in the *allCoreShell* mixing-state assumption (Figure 2d), the BC core diameter is further reduced to 86 nm and the shell thickness increased to 82 nm.

## 3 Results

### 3.1 Measurements of coating thickness as a function of rBC core size and fraction of rBC-containing particles

Figure 3 shows measured coating thickness as a function of the volume-equivalent diameter (VED) of the rBC cores (both the rBC cores and the coating are assumed to be spherical). The alternate axis gives the fraction of detectable notch positions (see Section 2.2.4 for details). The black dots represent the median and the shaded regions represent the interquartile range and 10th-90th percentile range of measurements for all particles across the different flights and altitudes. For rBC core diameters ranging from 140-220 nm (the region with greater than 90% successful fits), the median measured

coating thickness decreases slightly from 45 nm to 40 nm (with an interquartile range of: 30-70 nm to 17-65 nm). This results in total particle to rBC core diameter ratios ranging from 1.6 (IQR: 1.4-2.0) for rBC cores at 140 nm to 1.4 (IQR: 1.2-1.6) for rBC cores at 220 nm. This range is similar to measurements in the Canadian Arctic by Kondo et al. (2011). When combining with model results, we use only the measured core and shell thicknesses in the range with greater than 90% detectable notch positions (i.e. core diameters larger than 140 nm), and use minimum and maximum bounding assumptions

in the region 70-140 nm. Figure S1 shows the minimum and maximum shell thickness across the range 70-700 nm. A detailed examination of vertical variability in rBC measurements from this campaign is included in Schulz et al. (in prep).



Figure 4a shows the measured number distributions for uncoated rBC, coated rBC, and total particle number, with sets of symbols representing the average across an individual flight. The diameters of the coated rBC particles represent a sum of the rBC core diameter and twice the shell thickness. The total aerosol size distribution has a mode centered at near 150 nm, similar to measured accumulation-mode particles in the Arctic spring reported in Croft et al. (2016). The bare rBC size distribution peaks below the 70 nm size limit of the SP2. To estimate the size-resolved fraction of particles containing rBC, we take the ratio of the SP2-coated rBC particles and the UHSAS total particle size distribution (Figure 3b). In the median across all flights, we find that 16% of particles in the 175-330 nm diameter size range contain rBC, with values ranging from 10-29%. For particles with diameters ranging between 550-730 nm, this fraction decreases to 3%. This result is similar in magnitude with surface measurements of rBC-containing particle fractions from Sharma et al. (2017) from Alert, Canada. In general, we expect chemically aged plumes to be more internally mixed (through condensation and coagulation). The relatively low fractions of rBC-containing particles measured here may suggest local sources of non-BC accumulation-mode particles, such as sea salt, which is common in April at Alert (e.g. Leaitch et al., 2017), or slow coagulation timescales.

### 3.2 Comparison to model simulations

Figure 5 shows the size-resolved fraction of BC mass relative to total aerosol mass based on the ratio of the SP2 to USHAS compared to GEOS-Chem-TOMAS simulations. Due to instrument constraints, the measured size range is restricted to diameters of 175-730 nm compared to the TOMAS size range of 1 nm to 10 μm. Across the measurement size range, TOMAS predicts a lower BC mass fraction relative to total aerosol mass except at higher altitudes (~720 hPa) for diameters of 700 nm. The lower BC mass fraction is a result of both lower BC mass concentrations (consistent with previous studies, e.g. Xu et al., 2017) and higher non-BC mass simulated in GEOS-Chem-TOMAS than observed in the SP2 and UHSAS. In TOMAS, BC mass fraction is greatest at the surface for sub-200 nm diameter particles with a second peak at higher altitudes for particle diameters around 700 nm.

As the ratio of BC mass to total aerosol mass in TOMAS is lower than in measurements, the two different mixing-state constraints ($r_{shell}$-constrained and $fBC$-constrained) do not converge with the measurements. The effect of this is shown in Figure 6. In Figure 6a (left panel), we constrain all gridcells in the Arctic to have the measured fraction of BC-containing particles (shown in Figure 4b and re-binned to the TOMAS size bins in the black line in Figure 6b), and calculate the resulting BC core diameter and shell thickness. The black line in Figure 6a presents the measured core diameter and shell thickness for comparison. As the relative BC mass to total aerosol mass in TOMAS is less than observed (Figure 5), this results in smaller BC core diameters and larger shell thicknesses. Similarly, in Figure 6b, we constrain TOMAS aerosol mass to have the measured BC core diameters and shell thickness (shown in Figure 3 and re-binned to TOMAS sizes in the black line in Figure a), and calculate the resulting fraction of BC-containing particles. The resulting fraction of BC-containing particles for diameters of 200 nm using the $r_{shell}$-constrained method is a factor of 6 less than observed.



### 3.3 Implications for the direct radiative effect

To explore the impact of mixing state on DRE in the Arctic spring, we calculate the DRE due to all aerosol with the bounding mixing-state assumptions and the measurement constraints on mixing state. Figure 7 shows the net DRE over the Arctic for April assuming all BC is externally mixed. The pan-Arctic average DRE is -1.65 W m$^{-2}$, but locally there are

regions of opposing sign. Regions with a slight positive DRE are over areas where the aerosol mixture is darker than the underlying albedo. This occurs mostly over snow-covered land surfaces as opposed to oceans where the water and sea-ice has a lower albedo and the aerosol mixture is more reflective than the surface. We include a map of monthly mean albedo for the Arctic in April in the SI. We note that this result is likely sensitive to monthly average albedo estimates.

Table 3 presents the April, pan-Arctic mean DRE from all aerosol for the 5 different mixing-state assumptions. As

expected, the DRE calculated with the *external* mixing-state assumption produces the most negative DRE (-1.65 W m$^{-2}$), while the *allCoreShell* assumption produces the more positive DRE (-1.34 W m$^{-2}$). The *external\*1.5* assumption, included as a reference, is close to halfway (-1.52 W m$^{-2}$) between the two bounding assumptions. The *fBC-constrained* mixing state is less negative (-1.45 W m$^{-2}$) than the $r_{shell}$-*constrained* mixing state (-1.59 W m$^{-2}$). This is because constraining by the fraction of BC-containing particles results in smaller BC cores with larger shell thickness in TOMAS (Figure 6a) that leads to a

larger amount of absorption.

The difference in the DRE between the measurement-constrained mixing states (*fBC-constrained* and $r_{shell}$-*constrained*) and the bounding mixing states (*external*, *external\*1.5*, and *allCoreShell*) are plotted in Figure 8. Both the measurement-constrained mixing states are more positive than completely externally mixed BC. While less than 15% of particles contain BC, the *fBC-constrained* mixing state is 0.19 W m$^{-2}$ more positive than the *external* mixing-state

assumption (or 58% of the difference between *external* and *allCoreShell)* and 0.05 W m$^{-2}$ more positive than the *external\*1.5* mixing-state assumption (but 0.11 W m$^{-2}$ more negative that *allCoreShell* mixing-state assumption). Conversely, the $r_{shell}$-*constrained* mixing state is only 0.06 W m$^{-2}$ more positive than the *external* mixing-state assumption (or 20% of the difference between *external* and *allCoreShell*) and 0.07 W m$^{-2}$ less negative than the *external\*1.5* mixing-state assumption (but 0.25 W m$^{-2}$ more negative than the *allCoreShell* mixing-state assumption). Regionally the areas with

the largest differences between mixing states are over the areas with a slight positive net DRE in Figure 8 (Greenland, the Canadian high Arctic, and Russia). This further underscores the sensitivity to underlying surface albedo. In these regions, the *fBC-constrained* DRE is 0.3-0.4 W m$^{-2}$ more positive than the *external\*1.5* mixing-state assumption, while the $r_{shell}$-*constrained* mixing state is 0.3-0.4 W m$^{-2}$ more negative.

Table S1 presents the pan-Arctic mean DRE for the $r_{shell}$-*constrained* and *fBC-constrained* mixing states using the

minimum and maximum estimates for coating thickness for rBC core diameters less than 140 nm. In the pan-Arctic mean, the difference in DRE is less than 1% across the minimum and maximum bounding assumptions. This is likely caused by an overall minor contribution to optical extinction in particle sizes less than 140 nm.




### 3.4 Study limitations

We acknowledge several limitations in understanding the difference in DRE when using measurement constraints (instrument limitations are discussed in Section 2). First, we assume a core-shell morphology with BC at the exact center of the particle. Several studies have suggested this may not always be representative of atmospheric aerosol (e.g. Cappa et al.,

2012). We note that as we assume BC is always at the center of a mixed particle, BC absorption in this work is therefore an upper bound. Second, we assume all BC particles are coated (though sometimes with very thin coats as in the minimum coating assumption, see Supplemental Information). Some studies show that plumes may have a combination of coated and uncoated BC, with increasing in coated BC fraction increasing with chemical age of the plume (e.g. Subramanian et al., 2010). Third, we use averages of measurements across flights and altitudes for BC size distribution and coating information.

A separate paper, Schulz et al. (in prep), will examine spatial and vertical variability of BC measurements made in this campaign. Finally, the measurements used here are limited spatially and temporally. Future work may expand on the measurements reported here for other months and a larger spatial domain.

### 4. Conclusions

In this study, we present measurements of BC mixing state in the springtime Canadian high Arctic. Using an SP2

aboard the POLAR6 flights, we find that on average for rBC core diameters in the range of 140-220 nm, median coating thickness decreases from 45-40 nm (with an interquartile range of 30-70 nm to 17-65 nm). The ratio of total particle to rBC core diameter in this study (1.6-1.4) is comparable to measurements in the springtime Canadian Arctic by Kondo et al. (2011) and Sharma et al. (2017), as well as measurements in the European Arctic from Raatikainen et al. (2015) and Liu et al. (2015). Combining the SP2 size-resolved rBC measurements with total aerosol size distributions from the UHSAS

instrument, we estimate approximately 16% of particles contain rBC in the 175-330 nm diameter range, and 3% of particles contain rBC in the 550-730 nm diameter range. We use these measurements separately as constraints on BC mixing state simulated in TOMAS. When constraining TOMAS mass and number concentrations with the shell thickness as a function of rBC core diameter ($r_{shell}$-constrained), we calculate only 5% of 200 nm particles contain BC (compared to 16% from the SP2 and UHSAS observations). Conversely, constraining by the fraction of particles containing rBC (fBC-constrained), we find

the ratio of total particle to core diameter range from 2.6-2.7 across the SP2 size range (compared to 1.6 to 1.4). The reason these constraints do not converge towards the measurements is that GEOS-Chem-TOMAS simulates a lower BC mass concentration and higher non-BC mass concentration than measured by the SP2 and UHSAS, resulting in smaller BC mass to total aerosol mass ratio.

We estimate the Arctic DRE in April 2015 using bounding mixing-state assumptions of entirely externally mixed

and internally mixed BC and compare this to the DRE estimate using the measurement-constrained mixing states. We find that the fBC-constrained mixing state is 0.19 W m$^{-2}$ less negative than the completely external mixing-state assumption (and 0.11 more negative than the completely internal assumption), while the $r_{shell}$-constrained mixing state is 0.06 W m$^{-2}$ less



negative than the completely external mixing-state assumption (and 0.25 W m$^{-2}$ more negative than the completely internal assumption). As the BC mass fraction in TOMAS is lower than observed, constraining by the fraction of BC-containing particles results in smaller BC cores with larger shell thickness. Over regions of bright underlying surface albedo the differences can be greater than 0.4 W m$^{-2}$. The difference in the calculated DRE between $r_{shell}$-*constrained* and *fBC-*

*constrained* highlight the importance of accurately simulating all aerosol components in order to represent the chemical mixing state. As the focus of this study was on the Arctic, we recommend future work to focus on using this approach globally.

**Data Availability**

NETCARE (Network on Climate and Aerosols, 2015, http:// www.netcare-project.ca), which organized the aircraft flight

described in this paper, is moving towards a publicly available, online data archive. In the meantime, the data can be accessed by contacting the principal investigator of the network: Jon Abbatt at the University of Toronto (jabbatt@chem.utoronto.ca). In addition, the measurements of coating thickness and fraction of particles containing rBC are included as supplemental CSV files to this manuscript.

**Acknowledgements**

The authors acknowledge the financial support provided for NETCARE through the Climate Change and Atmospheric Research Program at NSERC Canada, as well as support from Environment and Climate Change Canada and the Alfred Wegener Institute. This research has also been supported by a grant from the U.S. Environmental Protection Agency's Science to Achieve Results (STAR) program through grant no. 83543801 and the U.S National Oceanic and Atmospheric Administration, an Office of Science, Office of Atmospheric Chemistry, Carbon Cycle, and Climate Program, under the

cooperative agreement award #NA17OAR430001. Finally, this material is based upon work supported by the National Science Foundation under Grant No. AGS-1559607.

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



**Table 1 Time, location, and duration of the portion of POLAR6 flights used in this study and shown in Figure 1**

| Period | Flight | Date | Takeoff/Landing location | Takeoff (UTC) | Landing (UTC) | Flight Time |
|---|---|---|---|---|---|---|
| Spring, 2015 | 2 | 7.4.2015 | Alert (82.5° N, 62.3° W) | 16:31:57 | 20:48:12 | 4:16:15 |
| | 3 | 8.4.2015 | Alert (82.5° N, 62.3° W) | 13:51:19 | 16:43:44 | 2:52:25 |
| | 4 | 8.4.2015 | Alert (82.5° N, 62.3° W) | 17:53:04 | 21:22:43 | 3:29:39 |
| | 5 | 9.4.2015 | Alert (82.5° N, 62.3° W) | 13:50:12 | 17:47:34 | 3:57:22 |
| | 6 | 11.4.2015 | Eureka (80.0° N, 85.9° W) | 15:57:28 | 21:16:05 | 5:18:37 |
| | 7 | 13.4.2015 | Eureka (80.0° N, 85.9° W) | 15:14:27 | 20:52:05 | 5:37:38 |



**Table 2. Description of BC mixing states**

| Mixing State | Description |
|---|---|
| *external* | All BC mass in a size range forms a separate particle from the other aerosol species with the same diameter. This mixing-state assumption provides a lower bound on absorption due to altering chemical mixing state. |
| $r_{shell}$-constrained | All BC mass in a size range forms the core of a mixed particle with measured rBC volume-equivalent diameter from the SP2. Hydrophilic scattering mass forms a shell around the core with measured shell thickness. Any remaining scattering mass forms a separate particle. |
| *external*1.5* | Identical to the *external* mixing-state assumption, but with BC absorption multiplied by a constant factor of 1.5 to simulate enhanced absorption in an external mixture, following the recommendation in Bond et al. (2006). This mixing-state assumption serves as an approximate "mid" point. |
| *fBC-constrained* | Simulated mass and number concentrations in a size range are split into a BC-containing and BC-free particle populations based on the measured rBC-containing particle fraction from the SP2 and UHSAS instruments. In the BC-containing population, core-shell morphologies are formed with diameters calculated based on the average BC mass or scattering mass per particle (calculated from the mass-to-number ratio). |
| *allCoreShell* | All BC mass in a size range forms the core of a particle with a concentric scattering shell. The total particle diameter is the sum of the core plus twice the shell thickness (and thus the BC diameter is less than the total diameter). This mixing-state assumption provides an upper bound on absorption due to altering chemical mixing state. |





**Table 3. The net direct radiative effect (DRE) calculated using each mixing state averaged over the Arctic for April.**

| Mixing-state assumption | Net DRE (W m$^{-2}$) |
|---|---|
| *external* | -1.65 |
| $r_{shell}$-*constrained* | -1.59 |
| *external*1.5* | -1.52 |
| *fBC-constrained* | -1.45 |
| *allCoreShell* | -1.34 |



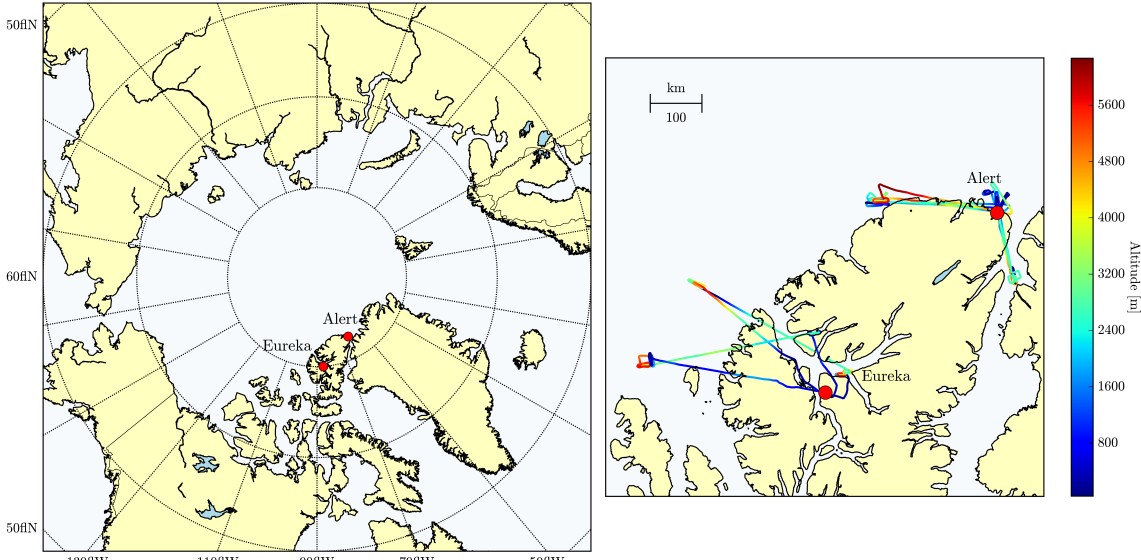

**Figure 1. Map showing the location of Alert and Eureka as well as the flight paths for the six flights in the Canadian High Arctic portion of the POLAR6 campaign. The flights were undertaken between the 7th and 13th of April 2015.**





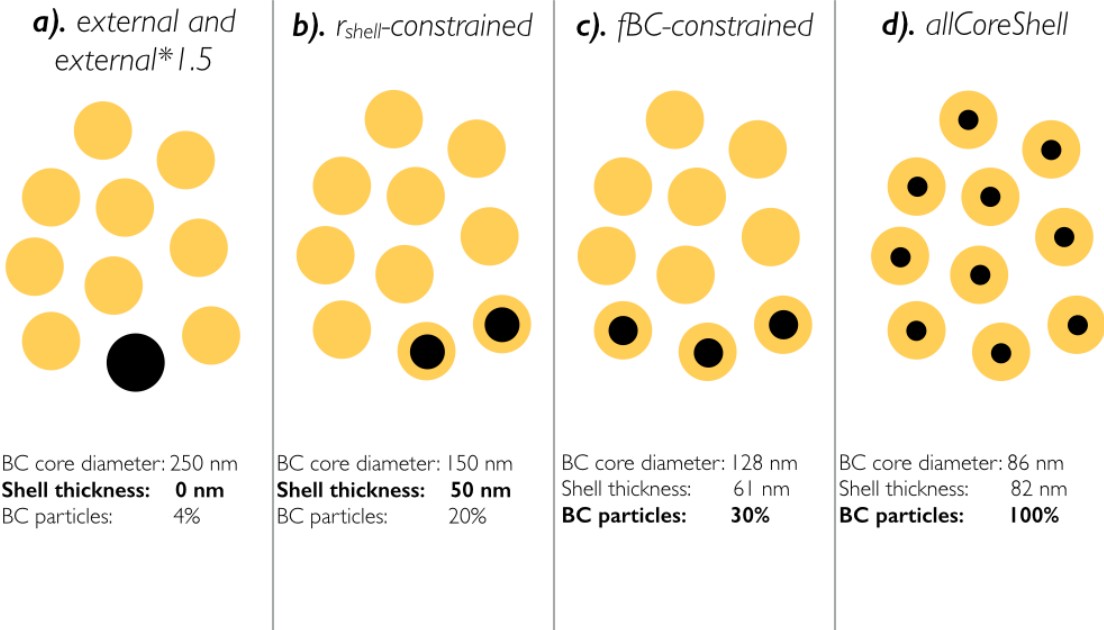

**Figure 2. Schematic presenting the different BC mixing states for the TOMAS size bin corresponding to particle diameters of 250 nm. The bold text shows the parameter being constrained in each mixing state. In this example, GEOS-Chem-TOMAS simulates 4% BC mass fraction and a particle number concentration of 100 cm$^{-3}$. In the *external* mixing-state assumption, all BC mass forms separate particles (rounded to 1 particle out of 10 for convenience), while in the *allCoreShell* mixing-state assumption, all BC mass is spread among all particles. The $r_{shell}$-constrained mixing state uses SP2 measurements of BC core diameter and shell thickness to constrain BC mass. The *fBC-constrained* mixing state uses BC-containing particle fractions from the SP2 and UHAS as the constraint on mixing state.**





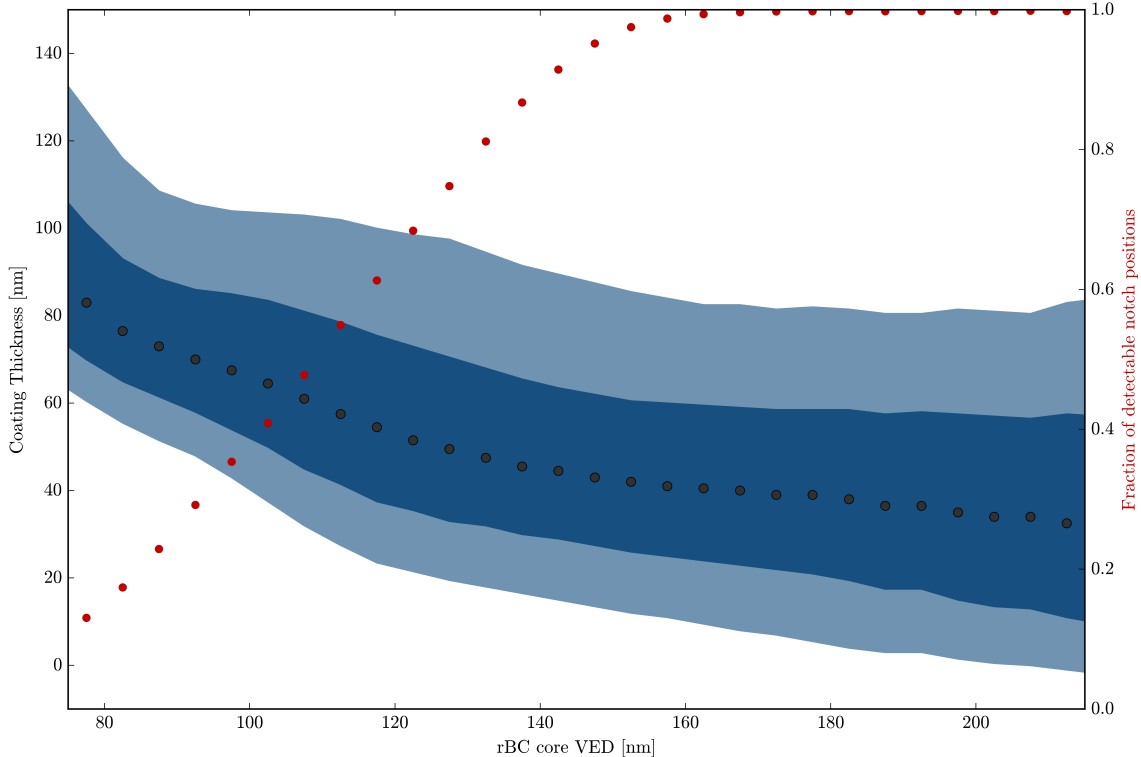

**Figure 3. Coating thickness as a function of rBC core diameter. Grey markers are the median, dark shaded area is the 25th-75 percentile, lighter shaded area is the 10-90th percentile of coating thicknesses for each bin.**



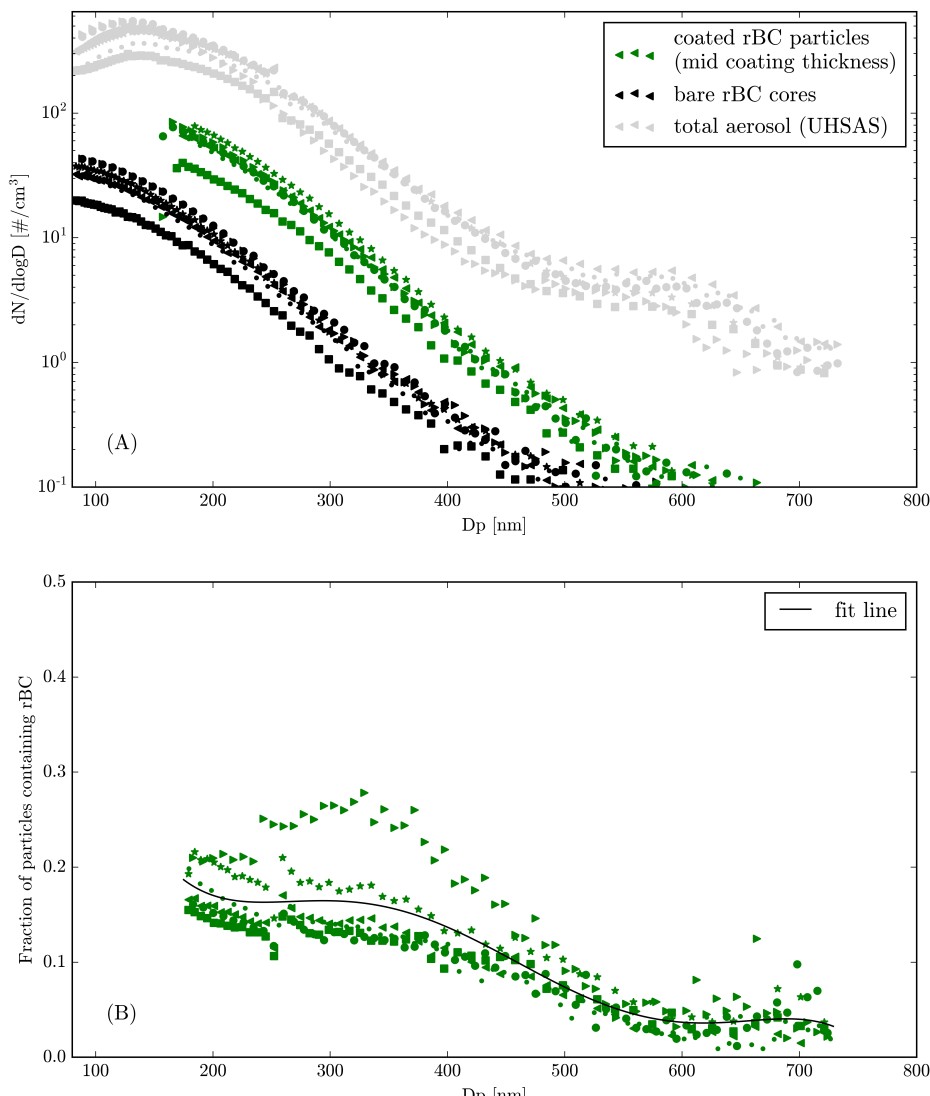

**Figure 4.** Number distributions for the bare rBC cores (black), the coated rBC-containing particles (green), and the total aerosol as detected by the UHSAS (grey) for the median coating thickness scenario. Shown in the bottom panel is the distribution of the fraction of the total aerosol containing rBC in the median coating scenario.



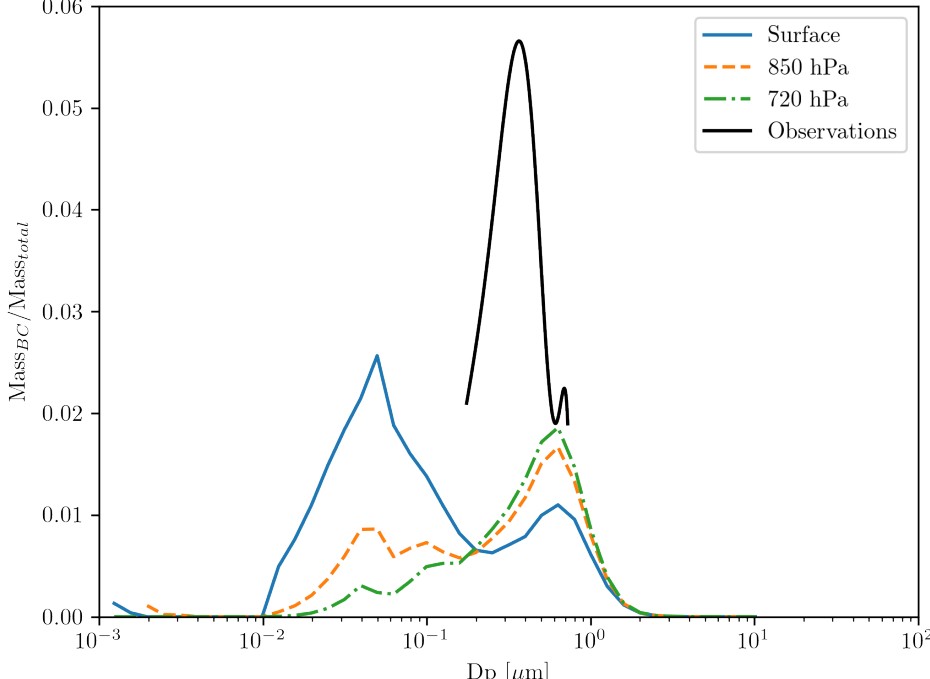

**Figure 5.** The fraction of BC aerosol mass relative total aerosol mass at each size bin in GEOS-Chem-TOMAS at three vertical levels along with the fraction determined from the SP2 and UHSAS observations.





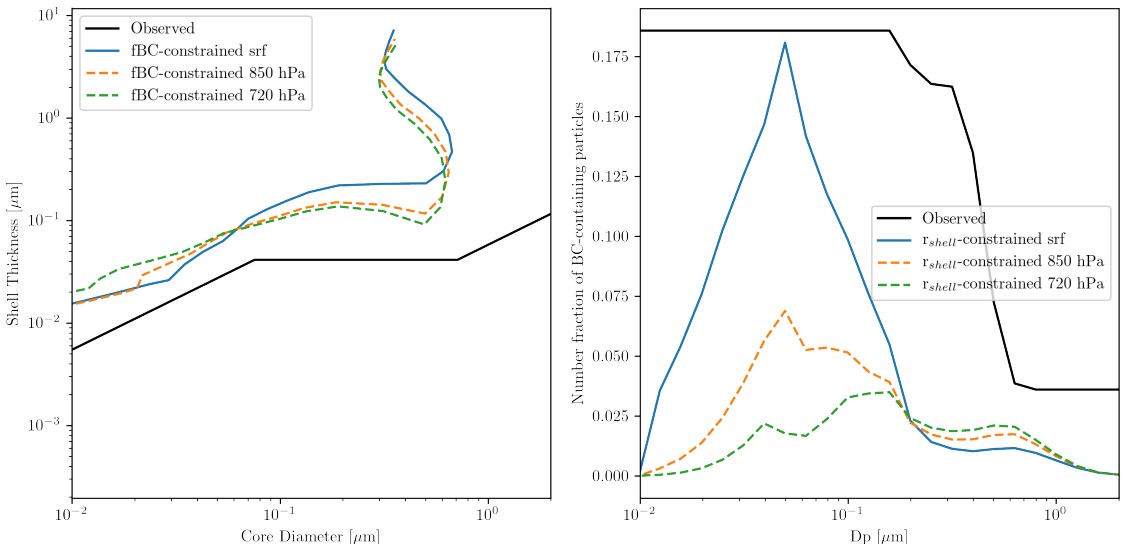

**Figure 6.** The simulated pan-Arctic mean shell thickness as a function of BC core diameter when the number fraction of BC is constrained by observations (left), and the simulated number fraction of BC when shell thickness as a function of core diameter is constrained by observations (right).



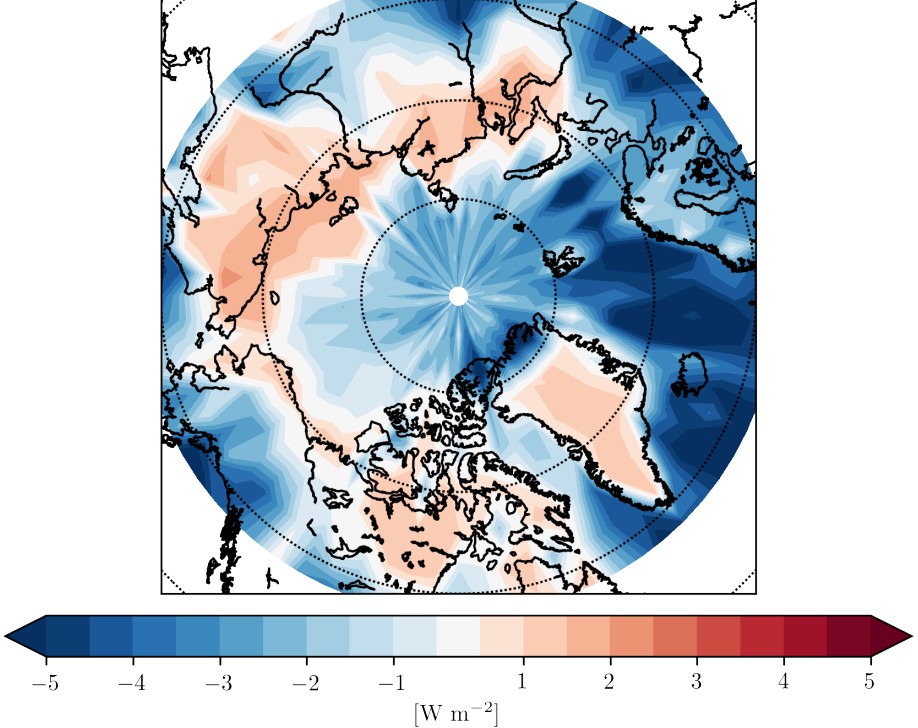

**Figure 7.** The net direct radiative effect over the Arctic in April assuming externally mixed particles.



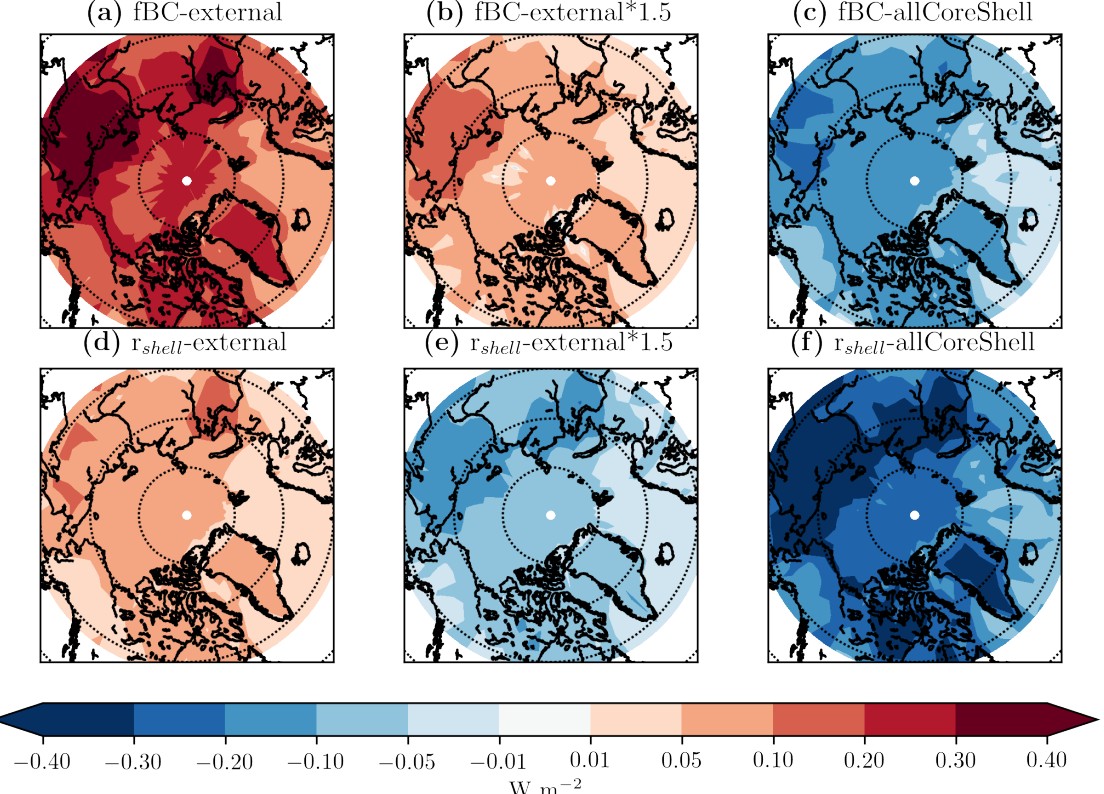

**Figure 8. The difference in DRE between _fBC-constrained_ (a,b,c) and _r_$_{shell}$_-constrained_ (d,e,f) compared to the bounding mixing-state assumptions.**

