# Peer review of "Size-resolved mixing state of black carbon in the Canadian high Arctic and implications for simulated direct radiative effect"

_Atmospheric Chemistry and Physics, 2018_

## Referee Comment (RC1) · Anonymous Referee #1 · 23 Mar 2018

In this paper, the authors present measurements and modeling of BC mixing state in the springtime Canadian high Arctic. Measurements were collected using Single Particle Soot Photometer (SP2) and Ultra-High Sensitivity Aerosol Spectrometer (UHSAS). The authors reported that measurements of aerosol mixing state provide important constraints for model estimates of direct radiative effect. The dataset and the associated analysis and modeling results are valuable to atmospheric and environmental researchers and the topic is fitting well within the scope of ACP. I suggest some revisions to improve the clarity and scientific merit of the current manuscript, after which I recommend publication.

[Figure]

Specific comments: 1. Particles measured by SP2 and UHSAS were combined to determine the number fraction of rBC particle. These two instruments have different size-cut and different measurement sensitivity. It would be worthwhile to include some details on the influence of the instrumental size-cut on the overall results, instrumental inter-calibration and any size adjustment for combining data from two different instruments. For example, if we have a 100 nm particle (mobility or some sort of equivalent size), do the SP2 and UHSAS both measure it as the same size. If not, how do the authors combine the measurements from two instruments to construct the distribution of the number fraction of rBC particles as a function of size? For the size distribution shown in Fig. 4A, how much are we missing below the measurement window (100 nm)? Is it possible to get a closure in Fig4A (i.e., total particles = bare-BC+ coated-BC+ non-BC)? They have some limited discussions on these, but it's hard to follow and get the whole picture. Details would be helpful for general readers. Details can go to the SI.

2. The discussion in Sec. 3.1 is limited. The authors should consider expanding the discussion in this section. There is no mention of Fig. 4B. What's the implication of fitted line in Fig4B? Can the authors propose any parameterization based on this fitting in Fig.4B, which can be used for constraining/evaluating model results in a similar environment where there are no measurements? The campaign-average mixing state is mostly focused in this section. Were there any changes in mixing states depending on air mass trajectory (relatively-fresh vs. aged) in that very clean environment?

3. Details measurements of aerosol properties are essential to improve model predictions and provide better constraints on the model results. However, based on the discussions in Sec. 3.3., it is not clear how much constraints are getting added by the detailed size-resolved BC mixing state measurements. It is clear that fully-externally or fully-internally mixed assumptions are not very realistic ones. However, external*1.5 bounding case vs. two constrained cases (fBC-constrained and rshell-constrained) showed a similar level of uncertainty on the estimated DRE. For example, In page 12,

L-26-28: "the fBC-constrained DRE is 0.3-0.4 W m-2 more positive than the external*1.5 mixing-state assumption, while the rshell constrained mixing state is 0.3-0.4 W m-2 more negative". Here, the two constraints cases provide two different results. If we don't know which measured case is a better representative one, then the details measurements are not adding that much of additional values compared to some bounding case. It would be worthwhile to include some discussions on this and current limitations and future directions that should be focused more.

4. The denominator of Eq. 1 [(r_core3)/(r_shell3+r_core3)] is confusing. If I understand correctly, to get the total volume of a coated-BC particle, we need to add core volume (r_core3) and shell volume (r_total3 - r_core3), where r_total= r_core+2*r_shell. In that case, [r_shell3+r_core3] would not provide the total volume of a coted-BC particle.

---

## Referee Comment (RC2) · Anonymous Referee #2 · 17 Apr 2018

The manuscript presents a case study examining radiative forcing by black carbon in the Arctic and its sensitivity to assumed mixing state. It uses observations made by an SP2 (black carbon and associated coatings) and a UHSAS (all aerosol) to constrain mixing state applied in the GEOS-Chem-TOMAS model and examines response of the calculated direct radiative effect. They find different mixing state assumptions lead to differences in DRE on the order of 0.3 W m-2, with observationally-constrained values falling within two bounding cases (complete external versus internal mixtures). The analysis is thorough and well within the scope of ACP, and will be of value to the community. I recommend its publication once the following minor points have been resolved.

[Figure]

General comment Both instruments used in this study are optically based, so they have a refractive index dependence. There should be a little more discussion on the uncertainties related to converting the optical measurements to a size comparable with the model that will be used in the DRE calculations. Further, an assumed RI is provided for the SP2 coating analysis. Could the UHSAS measurements be adjusted to have an RI consistent with this assumption (e.g., 1.5 for coating species applied to all non-rBC containing particles?

Specific comments Page 5, 16-22 - Were any laboratory tests performed to verify the lower limit for detection or is this number based on literature values?

Page 7 - some mention of refractive index impacts on sizing for both instruments (and a statement regarding how consistent the refractive index assumptions are between the two instruments (SP2 and UHSAS) is needed here.

Section 3.4 - While it may be obvious to most readers, I think it is worth pointing out the limitation of coating information being available only for a subset of the BC particles in this section in addition to the other limitations listed.

Figure 2 - I was a little confused by the wording in the caption: "rshell-constrained mixing state used SP2 measurements of BC core diameter and shell thickness to constrain BC mass". My understanding was that BC mass was always taken from the TOMAS simulation output. Does the caption mean BC mass per particle? Please clarify.

Figure 5 caption - should be "fraction of BC aerosol mass relative to total aerosol mass"? Also, could you provide an "average level" for the observations based on typical flight levels during the study? Same for Figure 6 as well?
* * *

---

## Referee Comment (RC3) · Anonymous Referee #3 · 28 Apr 2018

**General comments**

This manuscript describes using SP2 measurements of black carbon aerosol and its mixing state to constrain model predictions of direct radiative forcing in the Arctic region. The methods employed seem to be fairly unique; but I wonder if the results produced are valuable. Basically, there are two separate model runs tested (with appropriate base cases). One run constrains the coating thicknesses on black carbon aerosol with SP2 measurements while allowing the total mass of black carbon to be adjusted to whatever the model simulates. This results in fewer particles containing BC in the model, because the model predicts a smaller mass of BC than the SP2 mea-

surements do but larger non-BC mass. In the second model run, the fraction of BC containing particles relative to all particles is constrained by SP2/UHSAS measurements while the coating thicknesses are allowed to be adjusted to whatever the model simulates. This results in thicker coated BC particles because the model predicts more non-BC mass than the measurements show. My major concern with the manuscript is what does this actually tell us? If the magnitude of BC and non-BC aerosol is 'fixed' in the model (to match observations), either through improved emissions inventories or better transport, scavenging, etc., would that make both of these model runs more closely match each other? If the model isn't getting BC measurements right in any sense (mass or mixing state), then why is constraining just one of these at a time useful? Why not constrain both to the measurements?

What is the direct radiative forcing if the model is constrained to both BC mass and coating thickness as measured by the SP2? The answer to that relative to the base modeling case might be more useful than the two model runs described here.

A big assumption made in this work is that data from 7 flights over 1 week can be averaged and used as a monthly mean for the whole Arctic region. This is a big assumption that is not fully justified. The flights do not cover a significant region of the Arctic, so where do you get confidence that a campaign average of BC mass and coating thickness is valid for the whole Arctic for the whole month? How uniform are the SP2 measurements along the transects of each flight? Does the spatial variability in SP2 measurements match at all to the variability of the base model case in Figure 7? (It might be helpful to zoom in on the model to cover the flight region, which is very small on this map.) I think it is important to prove that this type of averaging gives useful data.

The manuscript is very well written and is a nice presentation of what was done. However, there is very little analysis of what was done. The work here needs to be put in context with other modeling methods and other observations. Spend some time telling me what these results mean.

**Specific comments**

The last 2 sentences of the Abstract confuse me a bit - if the measurements of mixing state are so important then why do the differences in the methods seem to be entirely from an underestimation of BC mass fraction?

Because not much is really said about the field mission as a whole, I wonder if Table 1 and Figure 1 could be moved to Supplemental?

Section 2.4 is a little confusing and I had to read it several times to fully understand it. I wonder if an illustration or schematic of some kind explaining this procedure should appear in the Supplemental? I don't think combining data from two SP2's in this way is common, so more explanation is warranted. Regarding this procedure, was the core-shell Mie model used to relate the core BC with coating to the scattering signals that the UHSAS measures? As described, it seems just diameter derived by the SP2 measurements was used, but this is not quite right - the light scattering signal is what should be compared.

Does the pattern of coating thicknesses in Fig 3 follow any specific functional form (e.g. diffusion-controlled growth laws)? Could a functional form be used to derive coating thickness over the whole range of BC cores in Fig 4a?

Fig 4, bottom panel, seems to show a single flight that was very different than the others. Can you explain this? If there is an outlier flight, is the average fit line really useful, or should the outlier be excluded?

Fig 5 Observations line has a weird squiggly part at the upper end - what is this?

The DRE numbers need some context. Are these significant changes? How do these number compare to other forcing mechanisms? Is BC the biggest forcer in the region?

The Conclusion states that this method should be applied globally, but I'm not yet convinced that this method provides any valuable insight. How do your model results compare to satellite remote sensing measurements? That would give an indication of

how your new modeling method matches the real world.

**Technical corrections**

Page 6 line 15 "and, and"

Page 7 line 2 "can results"

Page 11 line 6 should be "Figure 4b"

Page 13 line 32 "0.11" needs units
* * *

---

## Short Comment (SC1) · 28 Apr 2018

The authors used the SP2 and UHSAS measurements of BC mixing states to provide important observational constraints for model estimates of DRE over the Arctic. This work provides valuable information to advance the current understanding. I have a few comments/suggestions.

1. Previous observations (China et al., 2015) have shown a large variety of non-core-shell BC coating/mixing structures/morphology. Recent modeling studies (e.g., He et al., 2015, 2016) further indicated a substantial variation (up to a factor of 2) in BC optical properties by using those different coating/mixing structures/morphology. The

authors used a core-shell assumption for internally mixed BC, which may lead to some uncertainty. I suggest including these recent studies and add some discussions on this coating/mixing structure issue.

References:

China, S., et al.: Morphology and mixing state of aged soot particles at a remote marine free troposphere site: Implications for optical properties, Geophys. Res. Lett., 42, 1243–1250, doi:10.1002/2014gl062404, 2015.

He, C., et al.: Variation of the radiative properties during black carbon aging: theoretical and experimental intercomparison, Atmos. Chem. Phys., 15, 11967-11980, doi:10.5194/acp-15-11967-2015, 2015.

He, C., et al.: Intercomparison of the GOS approach, superposition T-matrix method, and laboratory measurements for black carbon optical properties during aging, J. Quant. Spectrosc. Radiat. Transf., 184, 287–296, doi:10.1016/j.jqsrt.2016.08.004, 2016.

2. Page 3, Lines 13-15: For the authors' information, a recent study (Qi et al., 2017a) also used the GEOS-Chem adjoint model to investigate the BC source contributions to the Arctic, which included most recent BC emissions (e.g., gas flaring, small-fire biomass burning, anthropogenic). This recent study is a useful reference that can be cited here.

References:

Qi, L., et al.: Sources of Springtime Surface Black Carbon in the Arctic: An Adjoint Analysis for April 2008, Atmos. Chem. Phys., 17, 9697-9716, doi:10.5194/acp-17-9697-2017, 2017a.

3. Section 2.5.1: recent studies (Qi et al., 2017b,c) have improved BC dry and wet deposition schemes in GEOS-Chem model, which leads to better BC simulations over the Arctic. The authors did not describe the BC dry and wet deposition scheme used in

the present study, which are key factors influencing BC simulations. I suggest adding some descriptions on BC deposition scheme used in the model and also including some discussions associated with the aforementioned recent improvements.

References:

Qi, L., et al.: Factors controlling black carbon distribution in the Arctic, Atmos. Chem. Phys., 17, 1037-1059, doi:10.5194/acp-17-1037-2017, 2017b.

Qi, L., et al.: Effects of the Wegener-Bergeron-Findeisen Process on Global Black Carbon Distribution, Atmos. Chem. Phys., 17, 7459-7479, doi:10.5194/acp-17-7459-2017, 2017c.

---

## Author Comment (AC1) · 21 Jun 2018

We thank Reviewer 1 for the constructive comments and suggestions. We have addressed all comments in our response and feel these revisions improve the manuscript. The reviewer comments are copied here in italics, our response is in plain text, and revised statements from the manuscript are underlined. We also submit a "marked up" version of the revised manuscript.

**Reviewer 1**

*In this paper, the authors present measurements and modeling of BC mixing state in the springtime Canadian high Arctic. Measurements were collected using Single Particle Soot Photometer (SP2) and Ultra-High Sensitivity Aerosol Spectrometer (UHSAS). The authors reported that measurements of aerosol mixing state provide important constraints for model estimates of direct radiative effect. The dataset and the associated analysis and modeling results are valuable to atmospheric and environmental researchers and the topic is fitting well within the scope of ACP. I suggest some revisions to improve the clarity and scientific merit of the current manuscript, after which I recommend publication.*

***Specific comments:***
*1. Particles measured by SP2 and UHSAS were combined to determine the number fraction of rBC particle. These two instruments have different size-cut and different measurement sensitivity. It would be worthwhile to include some details on the influence of the instrumental size-cut on the overall results, instrumental inter-calibration and any size adjustment for combining data from two different instruments. For example, if we have a 100 nm particle (mobility or some sort of equivalent size), do the SP2 and UHSAS both measure it as the same size. If not, how do the authors combine the measurements from two instruments to construct the distribution of the number fraction of rBC particles as a function of size? For the size distribution shown in Fig. 4A, how much are we missing below the measurement window (100 nm)? Is it possible to get a closure in Fig4A (i.e., total particles = bare-BC+ coatedBC+ non-BC)? They have some limited discussions on these, but it's hard to follow and get the whole picture. Details would be helpful for general readers. Details can go to the SI.*

Regarding instrumental inter-calibration and any size adjustment for combining data from two different instruments, both the SP2 and the UHSAS measure optical size using the scattered light intensity from an IR laser (1064nm light for the SP2 and 1054nm light for the UHSAS). To convert the scattering signal to a physical size, the refractive index of the particle is needed and the two instruments differ slightly in the refractive index used. To address the referee's comments, we have added additional information on the uncertainties in converting optical measurements to size in Sections 2.2.2 and 2.3.

Regarding the size cut, the detection ranges of the two instruments are given in Section 2.2.1 (SP2) and in Section 2.3 (UHSAS). Very few particles had rBC core diameters larger than 650nm, so the mismatch in detection range for large sizes is unlikely to be significant. If there

were particles with rBC cores smaller than the lower detection limit of the SP2 and very thick coatings, these would end up in the size range of the UHSAS but not be accounted for here as "rBC containing". However, we have tested a bounding case (See supplemental Figure S1) where small rBC components (including those below 75 nm) were assumed to have very thick coatings (double the median value). In this case, no substantial difference in the DRE was observed compared to the median coating-thickness case (see Supplemental section).

Revisions to the main text:
**In Section 2.2.2:** In the Mie model we used a refractive index of 2.26–1.26i for rBC (Moteki et al., 2010), which is appropriate for rBC from fossil fuel combustion. Other studies have used this value in analyses of particles from urban areas (Laborde et al. 2013) and from biomass burning (Taylor et al. 2014). In the Mie model we used a refractive index of 1.5-0.0i for the coating species. The value of 1.5-0.0i is appropriate for dry sulfate and sodium chloride (Schwarz et al., 2008a, 2008b). Previous studies have shown that varying the refractive index used for the rBC coating material from 1.45 to 1.59 had a minor impact on coating thickness (Taylor et al. 2014a).

**In Section 2.3:** Further details of the instrument operating principles can be found in Cai et al (2008). The UHSAS was calibrated using polystyrene latex spheres (PSLs), which have a refractive index of n=1.59. Kupc et al. (2018) investigated the size dependence of UHSAS measurements on the assumed refractive index of the particles being measured. For particles with diameters smaller than 600 nm and real refractive indices of n= 1.44–1.58, the diameter measured by the UHSAS varied by +4/−10% (relative to the refractive index of ammonium sulfate, n=1.52). For this study, a shift in the UHSAS size distribution by 10% would change the fraction of particles containing rBC by less than 3% which has a negligible impact on the DRE calculations.

**In Section 3.4:** Fourth, our measurements of BC mixing state only apply to BC particles in a limited size range. The size distributions in Figure 4 imply a substantial fraction of BC and non-BC number concentration exist outside the size range of our measurements. We attempt to account for this limitation in our model analysis through consideration of upper and lower bounds on coating and BC-containing particles.

*2. The discussion in Sec. 3.1 is limited. The authors should consider expanding the discussion in this section. There is no mention of Fig. 4B. What's the implication of fitted line in Fig4B? Can the authors propose any parameterization based on this fitting in Fig.4B, which can be used for constraining/evaluating model results in a similar environment where there are no measurements? The campaign-average mixing state is mostly focused in this section. Were there any changes in mixing states depending on air mass trajectory (relatively-fresh vs. aged) in that very clean environment?*

We have added several statements expanding on the measurements of BC mixing state during the POLAR6 campaign. We have copied these below, but first discuss the specific points brought up by the reviewer.

Thank you for catching the lack of reference to Figure 4b. We had mistakenly referenced a Figure 3b (which does not exist). We have corrected this error.

A parameterization based on the fit line in Figure 4b (or perhaps the results in Figure 3) is a really interesting idea; however, we feel it is beyond the scope of the current manuscript. The main reason is that it is not clear from the observations what underlying microphysical processes may control the fraction of BC containing particles (e.g., coagulation time scale, condensation rate, cloud processing). While it does not explitictly include physical processes, the fit line in Figure 4b is a polynomial fit that best matches the observations. As this may be helpful for future studies we have included the form of the fit in the caption to Figure 4b. In addition, we have added a couple sentences discussing the fitted line:

The solid line in Figure 4b is the best fit to the experimental data shown in Figure 4b. A polynomial function was used because this gave a good fit to the experimental data. The coefficients for this fit are provided in the caption to Figure 4; however, we note this fit is derived only from measurements in the springtime Arctic and is not based on any underlying understanding of the chemical or physical processes involved. With this polynomial fit, we…

And in the caption to Figure 4b:
The solid black line is a polynomial fit to the observations following the form (with x in nm): $y = 1.25 - (1.52*10^{-2})*x + (8.05*10^{-5})*x^2 - (2.00*10^{-7})*x^3 + (2.31*10^{-10})*x^4 - (1.00*10^{-13})*x^5$.

Overall, we did not find substantial variability in observed mixing state parameters across flights. This is likely due to relatively long transport (~days) of most aerosol sampled during the campaign. We have added the following statements:
Across all flights (and altitudes), we do not find substantial variability in measured coating thickness. This lack of variability is likely due to atmospheric processing over several days during transport of the air mass to the Arctic region. In the SI, we plot coating thickness for each flight in Figure S3. The low degree of variability across flights justifies the use of campaign averages in our model analysis.

*3. Details measurements of aerosol properties are essential to improve model predictions and provide better constraints on the model results. However, based on the discussions in Sec. 3.3., it is not clear how much constraints are getting added by the detailed size-resolved BC mixing state measurements. It is clear that fully-externally or fully-internally mixed assumptions are not very realistic ones. However, external\*1.5 bounding case vs. two constrained cases (fBC-*

*constrained and rshell-constrained) showed a similar level of uncertainty on the estimated DRE. For example, In page 12, L-26-28: "the fBC-constrained DRE is 0.3-0.4 W m-2 more positive than the external\*1.5 mixing-state assumption, while the rshell constrained mixing state is 0.3-0.4 W m-2 more negative". Here, the two constraints cases provide two different results. If we don't know which measured case is a better representative one, then the details measurements are not adding that much of additional values compared to some bounding case. It would be worthwhile to include some discussions on this and current limitations and future directions that should be focused more.*

This is an important point to make in our paper, and we have added several statements explicitly addressing this comment. While we are unable to fully constrain the model to observations of BC mixing state, we are able to reduce the range in estimates of the DRE due to BC mixing-state assumptions using these measurement constraints. For instance, the range of estimated DRE of the pan-Arctic mean using the fully internal/external bounding assumptions is 0.31 W m$^{-2}$. Using the measurement constraints, this range is reduced to 0.14 W m$^{-2}$. This reduction of a factor of two in the range of the DRE due to uncertainty in the mixing state of BC is an improvement over only using bounding mixing-state assumptions. The remaining range of 0.14 W m$^{-2}$ is due to the interaction of mixing state with the uncertainty in the ratio of scattering to BC mass. Thus, this study demonstrates that the uncertainty range in DRE can be reduced by only constraining BC mixing state (and not aerosol mass concentrations); however, this approach is limited to the representation of the scattering-to-BC mass ratio. If the model does simulate the observed scattering-to-BC mass ratio, than the uncertainty range in DRE would be further reduced.

To clarify this point, we have added the following statements:

***In the abstract:***
We find that the pan-Arctic average springtime DRE ranges from -1.65 W m$^{-2}$ to -1.34 W m$^{-2}$ when assuming entirely externally or internally mixed BC. This range in DRE is reduced by over a factor of two (-1.59 to -1.45 W m$^{-2}$) when using the observed mixing-state constraints.

***In Section 3.3:***
The range in the pan-Arctic mean DRE using the bounding mixing-state assumptions (*external, allCoreShell*) is 0.3 W m$^{-2}$ (or about 20%). Conversely, using the measurement constraints on BC mixing state ($r_{shell}$-*constrained, fBC-constrained*) results in a range of pan-Arctic mean DRE of 0.14 W m$^{-2}$ (or about 9%). Thus, using the measurement constraints on BC mixing state reduces the range in DRE from assumptions of BC mixing state by a factor of two.

***In Section 4:***

The estimated range of the DRE using the measurement-constrained mixing states (-1.45 to -1.59 W m$^{-2}$) is approximately a factor of two less than the estimated range when using the bounding mixing-state assumptions (-1.34 to -1.65 W m$^{-2}$).

If the BC mass fraction in TOMAS does match the observations, both mixing state parameters (shell thickness and fraction of BC-containing particles) can be constrained simultaneously and the DRE in both cases would be identical.

*4. The denominator of Eq. 1 [(r_core3)/(r_shell3+r_core3)] is confusing. If I understand correctly, to get the total volume of a coated-BC particle, we need to add core volume (r_core3) and shell volume (r_total3 - r_core3), where r_total= r_core+2\*r_shell. In that case, [r_shell3+r_core3] would not provide the total volume of a coted-BC particle.*

This is not quite accurate (though we appreciate where the confusion arises from). We are using the term r_shell to refer to the shell thickness (the distance from the outer edge of the core to the outer edge of the particle). The total radius of a coated particle is equal to simply the core radius plus the shell radius (r_total = r_core + r_shell) and not twice the shell radius. In other parts of the paper, we discuss the diameter of the coated particle, which is calculated as dp_total = d_core + 2\*r_shell. The "2\*r_shell" term is used here as the SP2 measures shell radius (often referred to as "shell thickness"). In addition, we feel discussing "shell diameter" may be misleading as it may not be clear to a reader if this term does or does not include the core diameter.

---

## Author Comment (AC2) · 21 Jun 2018

We thank Reviewer 2 for the constructive comments and suggestions. We have addressed all comments in our response and feel these revisions improve the manuscript. The reviewer comments are copied here in italics, our response is in plain text, and revised statements from the manuscript are underlined. We also submit a "marked up" version of the revised manuscript.

**Reviewer 2**

*The manuscript presents a case study examining radiative forcing by black carbon in the Arctic and its sensitivity to assumed mixing state. It uses observations made by an SP2 (black carbon and associated coatings) and a UHSAS (all aerosol) to constrain mixing state applied in the GEOS-Chem-TOMAS model and examines response of the calculated direct radiative effect. They find different mixing state assumptions lead to differences in DRE on the order of 0.3 W m-2, with observationally-constrained values falling within two bounding cases (complete external versus internal mixtures). The analysis is thorough and well within the scope of ACP, and will be of value to the community. I recommend its publication once the following minor points have been resolved.*

***General comment***

*Both instruments used in this study are optically based, so they have a refractive index dependence. There should be a little more discussion on the uncertainties related to converting the optical measurements to a size comparable with the model that will be used in the DRE calculations. Further, an assumed RI is provided for the SP2 coating analysis. Could the UHSAS measurements be adjusted to have an RI consistent with this assumption (e.g., 1.5 for coating species applied to all non-rBC containing particles?*

The referee raises important points here. To address the referee's comments on the uncertainties related to converting optical measurements to size, we have added additional information to Sections 2.2.2 and 2.3. In these sections, we discuss the relative impact of the refractive indices used for the rBC coating material and the UHSAS measurements. While these do affect the results, their impact is minor in comparison to the refractive index used for the rBC core.

Section 2.2.2, paragraph 2 has been modified to read:

With the scattering amplitude determined by the leading-edge technique, and the measured rBC core diameter, a core-shell Mie model can be used to determine the optical diameter of the rBC-containing particles. In the Mie model we used a refractive index of 2.26–1.26i for rBC (Moteki et al., 2010), which is appropriate for rBC from fossil fuel combustion. Other studies have used this value in analyses of particles from urban areas (Laborde et al. 2013) and from biomass burning (Taylor et al. 2014). In the Mie model we used a refractive index of 1.5-0.0i for the coating species. The value of 1.5-0.0i is appropriate for dry sulfate and sodium chloride (Schwarz et al., 2008a, 2008b). Previous studies have shown that varying the refractive index

used for the rBC coating material from 1.45 to 1.59 had a minor impact on coating thickness (Taylor et al. 2015).

The following has been added to Section 2.3, paragraph 1:

The UHSAS was calibrated using polystyrene latex spheres (PSLs), which have a refractive index of n=1.59. Kupc et al. (2018) investigated the size dependence of UHSAS measurements on the assumed refractive index of the particles being measured. For particles with diameters less than 600 nm and real refractive indices of n= 1.44–1.58, the diameter measured by the UHSAS varied by +4/−10% (relative to the refractive index of ammonium sulfate, n=1.52). For this study, although significant, a shift in the UHSAS size distribution by 10% would change the fraction of particles containing rBC by less than 3% which has a negligible impact on the DRE calculations.

*Specific comments*
*Page 5, 16-22 - Were any laboratory tests performed to verify the lower limit for detection or is this number based on literature values?*

We have added the following to address the reviewers comment.

Both SP2s were calibrated against size selected external BC standards. The lower limit for mass was set at a point where detection efficiency for particles compared to a CPC was close to 100%.

*Page 7 - some mention of refractive index impacts on sizing for both instruments (and a statement regarding how consistent the refractive index assumptions are between the two instruments (SP2 and UHSAS) is needed here.*

As discussed above, we have added statements discussing the refractive index used for the UHSAS to Section 2.3, and statements discussing the refractive indices used for the SP2 to Section 2.2.2.

*Section 3.4 - While it may be obvious to most readers, I think it is worth pointing out the limitation of coating information being available only for a subset of the BC particles in this section in addition to the other limitations listed.*

This is certainly a worthwhile point to emphasize. We have added the following statements to this section:

Fourth, our measurements of BC mixing state only apply to BC particles in a limited size range. The size distributions in Figure 4 imply a substantial fraction of BC and non-BC number concentration exist outside the size range of our measurements. We attempt to account for this limitation in our model analysis through consideration of upper and lower bounds on coating and BC-containing particles

*Figure 2 - I was a little confused by the wording in the caption: "rshell-constrained mixing state used SP2 measurements of BC core diameter and shell thickness to constrain BC mass". My understanding was that BC mass was always taken from the TOMAS simulation output. Does the caption mean BC mass per particle? Please clarify*

We agree this caption was worded poorly. Reviewer 2 is correct, we did intend for this to read "BC mass per particle"; however, this seems unnecessary and confusing. We have re-written this sentence to be more direct:
The $r_{shell}$-constrained mixing state uses SP2 measurements to constrain BC core diameter and shell thickness.

*Figure 5 caption - should be "fraction of BC aerosol mass relative to total aerosol mass"? Also, could you provide an "average level" for the observations based on typical flight levels during the study? Same for Figure 6 as well?*

Yes, thank you, we have corrected this sentence.

Overall, we do not find substantial vertical variability in our measurements of BC mixing state. The three model layers provided in Figures 5 and 6 is meant to show the respective comparisons across the relevant range of altitudes from the measurements (shown in Figure 1). To this end, we have replaced the 720 hPa layer with the model layer corresponding to 550 hPa. The vertical range from the surface to 550 hPa encompasses much of the altitude ranges from the POLAR6 flights. Given that the model does not overlap the observations over this range, we do not feel an average level is helpful in this instance. Further, this may mislead readers into thinking we sampled the model along the flight track which is not the case.

Revisions to manuscript:
In Figures 5 and 6, we replace the 720 hPa level with 550 hPa (see revised submission). We make minor changes to the text, replacing references to 720 hPa with 550 hPa.

---

## Author Comment (AC3) · 21 Jun 2018

We thank Reviewer 3 for the constructive comments and suggestions. We have addressed all comments in our response and feel these revisions improve the manuscript. The reviewer comments are copied here in italics, our response is in plain text, and revised statements from the manuscript are underlined. We also submit a "marked up" version of the revised manuscript.

**Reviewer 3**

*This manuscript describes using SP2 measurements of black carbon aerosol and its mixing state to constrain model predictions of direct radiative forcing in the Arctic region. The methods employed seem to be fairly unique; but I wonder if the results produced are valuable. Basically, there are two separate model runs tested (with appropriate base cases). One run constrains the coating thicknesses on black carbon aerosol with SP2 measurements while allowing the total mass of black carbon to be adjusted to whatever the model simulates. This results in fewer particles containing BC in the model, because the model predicts a smaller mass of BC than the SP2 measurements do but larger non-BC mass. In the second model run, the fraction of BC containing particles relative to all particles is constrained by SP2/UHSAS measurements while the coating thicknesses are allowed to be adjusted to whatever the model simulates. This results in thicker coated BC particles because the model predicts more non-BC mass than the measurements show. My major concern with the manuscript is what does this actually tell us? If the magnitude of BC and non-BC aerosol is 'fixed' in the model (to match observations), either through improved emissions inventories or better transport, scavenging, etc., would that make both of these model runs more closely match each other? If the model isn't getting BC measurements right in any sense (mass or mixing state), then why is constraining just one of these at a time useful? Why not constrain both to the measurements?*

We appreciate the reviewer's feedback. As it is important to directly communicate our research goals in this project, we have added several statements to the manuscript to better state the value of our results. These statements are copied below, and we will briefly elaborate on our goals here.

The main objective of this work is to explore the possibility of reducing the uncertainty range in model estimates of the direct radiative effect (DRE) by constraining only the BC mixing state through observations. Previous research has demonstrated that the mixing state of BC is a major uncertainty in estimating the DRE; however, many chemical-transport models (and climate models) do not explicitly simulate mixing state. Instead, modeling studies typically assume entirely internal or external mixing-state assumptions (for references and further discussion see the Introduction). To improve upon this assumption, we use observations of two population mixing-state parameters (coating thickness and fraction of particles containing BC). A main finding of this work is that when using measurement constraints, we reduce the range in estimated DRE by more than a factor of two relative to estimates of DRE using the bounding

mixing-state assumptions. The remaining range in DRE is due to model bias in the BC-to-scattering aerosol mass fraction (not the total mass concentration).

Of course, constraining both mass and mixing state would further improve model estimates of the DRE. However, it is not always possible to constrain aerosol mass concentrations everywhere. Aerosol mass concentrations tend to be quite variable spatially and temporally. Conversely, our measurements seem to indicate that BC mixing state is roughly constant throughout the Arctic in Spring (more on this in a later comment). As aerosol models do explicitly track aerosol mass concentration, we choose instead to study the possibility of constraining a parameter models do not typically track.

If the model does reproduce the observed BC-to-scattering mass *ratio* then the two mixing-state constraints would produce identical DRE estimates.

To summarize, this study accomplishes the following:
1. Presents measurements of BC mixing state in the Arctic spring.
2. Combines these measurements with model simulations to reduce the uncertainty range in DRE due to mixing state of BC by more than a factor of 2.
3. Demonstrates how the uncertainty in mixing state of BC is not entirely independent of the ratio of BC to non-BC mass (again the ratio not absolute concentrations).

Overall, this work presents a first step in constraining an important parameter in model estimates of the DRE  by combining measurements from a commonly used instrument (the SP2) with aerosol mass concentrations tracked in most models.

We have added the following statements to help address our response to this reviewer comment in the manuscript:

***In the abstract:***
We find that the pan-Arctic average springtime DRE ranges from -1.65 W m$^{-2}$ to -1.34 W m$^{-2}$ when assuming entirely externally or internally mixed BC. This range in DRE is reduced by over a factor of two (-1.59 to -1.45 W m$^{-2}$) when using the observed mixing-state constraints.

***In Section 3.3 (Results):***
The range in the pan-Arctic mean DRE using the bounding mixing-state assumptions (*external, allCoreShell*) is 0.3 W m$^{-2}$ (or about 20%). Conversely, using the measurement constraints on BC mixing state ($r_{shell}$-constrained, fBC-constrained) results in a range of pan-Arctic mean DRE of 0.14 W m$^{-2}$ (or about 9%). Thus, using the measurement constraints on BC mixing state reduces the range in DRE from assumptions of BC mixing state by a factor of two.

***In Section 4 (Conclusions):***
The estimated range of the DRE using the measurement-constrained mixing states (-1.45 to -1.59 W m$^{-2}$) is approximately a factor of two less than the estimated range when using the bounding mixing-state assumptions (-1.34 to -1.65 W m$^{-2}$).

*What is the direct radiative forcing if the model is constrained to both BC mass and coating thickness as measured by the SP2? The answer to that relative to the base modeling case might be more useful than the two model runs described here.*

We do not have sufficient observations to constrain BC mass concentration horizontally, vertically, and temporally throughout the Arctic spring. In general, we found that the mixing state of BC in the Arctic exhibit less variability than absolute BC mass concentrations (the latter will be discussed in detail in Schulz et al., 2018). Further, the DRE is also dependent on non-BC aerosol mass concentrations.

We do find in our work that constraining the BC mixing state in models is not entirely independent of the ratio of BC-to-scattering mass. If the model predicted the same ratio as observed, than the estimates of DRE using the two mixing-state constraints would be the same.

We have added the following sentences stating this:
**In Section 3.2**: If the BC mass to total aerosol mass ratio in GEOS-Chem-TOMAS matched the observations, both the shell thickness and fraction of BC-containing particles could be constrained simultaneously.

**In Section 4**:  If the BC mass fraction in TOMAS does match the observations, both mixing state parameters (shell thickness and fraction of BC-containing particles) can be constrained simultaneously and the DRE in both cases would be identical.

*A big assumption made in this work is that data from 7 flights over 1 week can be averaged and used as a monthly mean for the whole Arctic region. This is a big assumption that is not fully justified. The flights do not cover a significant region of the Arctic, so where do you get confidence that a campaign average of BC mass and coating thickness is valid for the whole Arctic for the whole month? How uniform are the SP2 measurements along the transects of each flight? Does the spatial variability in SP2 measurements match at all to the variability of the base model case in Figure 7? (It might be helpful to zoom in on the model to cover the flight region, which is very small on this map.) I think it is important to prove that this type of averaging gives useful data.*

Thank you for pointing this out. We have added several statements to justify this assumption. First, we would like to clarify that we do not take a campaign average BC mass concentration.

We use simulated BC (and other aerosol species) mass and number concentrations from GEOS-Chem-TOMAS. We do use average coating thickness across the campaign. Overall, we did not find substantial variability in observed coating thickness across flights. This is likely due to relatively long transport (~days) of most aerosol sampled during the campaign. We have added the following statements:

Across all flights (and altitudes), we do not find substantial variability in measured coating thickness. This lack of variability is likely due to atmospheric processing over several days during transport of the air mass to the Arctic region. In the SI, we plot coating thickness for each flight in Figure S3. The low degree of variability across flights justifies the use of campaign averages in our model analysis.

Figure 7 shows the DRE due to all aerosol species, and not only BC. Aerosol optical depth has a strong dependency on non-BC aerosol mass concentration, and so we do not expect variability in BC mass concentration to explain all of the variability in simulated DRE. As discussed in the main text, some of the variability seen in Figure 7 (notably the portions of positive DRE) are likely caused by variability in underlying surface albedo. This is explored further in the Supplemental Material.

*The manuscript is very well written and is a nice presentation of what was done. However, there is very little analysis of what was done. The work here needs to be put in context with other modeling methods and other observations. Spend some time telling me what these results mean.*

We feel we have added a number of statements further elaborating on our results. In addition, we make a number of comparisons to past observations in the Introduction, Section 3.1, and Section 4 of the main text. Copied below are several of the additions:

We find that the pan-Arctic average springtime DRE ranges from -1.65 W m$^{-2}$ to -1.34 W m$^{-2}$ when assuming entirely externally or internally mixed BC. This range in DRE is reduced by over a factor of two (-1.59 to -1.45 W m$^{-2}$) when using the observed mixing-state constraints.

The range in the pan-Arctic mean DRE using the bounding mixing-state assumptions (*external, allCoreShell*) is 0.3 W m$^{-2}$ (or about 20%). Conversely, using the measurement constraints on BC mixing state ($r_{shell}$-*constrained, fBC-constrained*) results in a range of pan-Arctic mean DRE of 0.14 W m$^{-2}$ (or about 9%). Thus, using the measurement constraints on BC mixing state reduces the range in DRE from assumptions of BC mixing state by a factor of two.

The estimated range of the DRE using the measurement-constrained mixing states (-1.45 to -1.59 W m$^{-2}$) is approximately a factor of two less than the estimated range when using the bounding mixing-state assumptions (-1.34 to -1.65 W m$^{-2}$).

If the BC mass to total aerosol mass ratio in GEOS-Chem-TOMAS matched the observations, both the shell thickness and fraction of BC-containing particles could be constrained simultaneously.

If the BC mass fraction in TOMAS does match the observations, both mixing state parameters (shell thickness and fraction of BC-containing particles) can be constrained simultaneously and the DRE in both cases would be identical.

*Specific comments*
*The last 2 sentences of the Abstract confuse me a bit - if the measurements of mixing state are so important then why do the differences in the methods seem to be entirely from an underestimation of BC mass fraction?*

We have re-written these sentences to better reflect our results. The measurement constraints on mixing state reduce the range in estimated DRE from only using bounding mixing-state assumptions by over a factor of two. The remaining range in DRE across the two measurement constrained mixing states is due to interaction with the uncertainty in the BC to total aerosol mass fraction.

Revised sentences:
We find that the pan-Arctic average springtime DRE ranges from -1.65 W m$^{-2}$ to -1.34 W m$^{-2}$ when assuming entirely externally or internally mixed BC. This range in DRE is reduced by over a factor of two (-1.59 to -1.45 W m$^{-2}$) when using the observed mixing-state constraints. The difference in DRE between the two observed mixing-state constraints is due to an underestimation of BC mass fraction in the springtime Arctic in GEOS-Chem-TOMAS compared to POLAR6 observations.

*Because not much is really said about the field mission as a whole, I wonder if Table 1 and Figure 1 could be moved to Supplemental?*

This is a fair suggestion. As the level of detail in Table 1 does not add to the paper, we have moved this to the Supplemental Material. However, as we do not feel that the paper has an excess of figures, we feel it is helpful to retain Figure 1 in the main text. Figure 1 provides context for the range of vertical levels shown in Figures 5 and 6.

Revisions:
The reference to Table 1 in the main text has been changed to Table S1 (Supplemental Table 1). In the Supplemental Material, we have added this Table with a short description.

*Section 2.4 is a little confusing and I had to read it several times to fully understand it. I wonder if an illustration or schematic of some kind explaining this procedure should appear in the Supplemental? I don't think combining data from two SP2's in this way is common, so more explanation is warranted. Regarding this procedure, was the core-shell Mie model used to relate the core BC with coating to the scattering signals that the UHSAS measures? As described, it seems just diameter derived by the SP2 measurements was used, but this is not quite right - the light scattering signal is what should be compared.*

The reviewer has raised a good point. We have added a section to the Supplemental material giving details and an example of how the data from the two SP2s and the UHSAS was combined. In this added section in the Supplemental, we provide enumerated steps to clearly demonstrate the procedure. Additionally, Supplemental Figure S1 provides a graphical example on this process.

*Does the pattern of coating thicknesses in Fig 3 follow any specific functional form (e.g. diffusion-controlled growth laws)? Could a functional form be used to derive coating thickness over the whole range of BC cores in Fig 4a?*

A parameterization based on the fit line in Figure 4b (or perhaps the results in Figure 3) is a really interesting idea; however, we feel it is beyond the scope of the current manuscript. The main reason is that it is not clear from the observations what underlying microphysical processes may control the fraction of BC containing particles (e.g., coagulation time scale, condensation rate, cloud processing), and these processes will influence the mixing state outside of the measurement-constrained size range. While it does not explitictly include physical processes, the fit line in Figure 4b is a polynomial fit that best matches the observations. As this may be helpful for future studies we have included the form of the fit in the caption to Figure 4b. We have added the following statements to the main text to clarify this:
The solid line in Figure 4b is the best fit to the experimental data shown in Figure 4b. A polynomial function was used because this gave a good fit to the experimental data. The coefficients for this fit are provided in the caption to Figure 4; however, we note this fit is derived only from measurements in the springtime Arctic and is not based on any underlying understanding of the chemical or physical processes involved. With this polynomial fit, we…

And in the caption to Figure 4b:
The solid black line is a polynomial fit to the observations following the form (with x in nm): $y = 1.25 - (1.52 \times 10^{-2}) \cdot x + (8.05 \times 10^{-5}) \cdot x^2 - (2.00 \times 10^{-7}) \cdot x^3 + (2.31 \times 10^{-10}) \cdot x^4 - (1.00 \times 10^{-13}) \cdot x^5$.

*Fig 4, bottom panel, seems to show a single flight that was very different than the others. Can you explain this? If there is an outlier flight, is the average fit line really useful, or should the outlier be excluded?*

There is not an obvious reason why this flight is different than the others. As we do not have a reason for the difference, we do not feel justified in removing it from the average.

*Fig 5 Observations line has a weird squiggly part at the upper end - what is this?*

This is most likely caused by a lower number of particle counts in this size range leading to a noiser fit.

*The DRE numbers need some context. Are these significant changes? How do these number compare to other forcing mechanisms? Is BC the biggest forcer in the region?*

In this work, we are really interested in the range of DRE due to uncertainty in BC mixing state. To add some context to the change in this range due to measurement constraints on mixing state, we have rephrased our results referring to a factor of two reduction in the range of estimated DRE (these statements have been shared in response to earlier comments). This reduction in uncertainty in DRE is the context we wish readers to take away from this study.

A factor of two reduction is certainty significant. A comparison of our simulated DRE in terms of absolute numbers is difficult, as there are not many studies providing a similar estimate of the direct radiative effect (as opposed to a direct radiative forcing). In this work, we follow the terminology described in Heald et al. (2014), where a forcing is relative to some time period (usually 1750 or 1850) and an effect is relative a case with no aerosol. We do not feel other effect mechanisms in the Arctic (such as the cloud effect, generally 10s of W m$^{-2}$) are relevant comparisons. Generally, a change in DRE on the order of 0.15 W m$^{-2}$ (as we see for mixing state) is about on the order we may expect to see in the global mean due to turning on and off a source sector (such as biomass burning). However, as we do not explore this in this work, we feel this would be a distracting comparison to make.

With regard to the question about BC being the biggest forcer in the region, in the first paragraph of this manuscript we cite studies suggesting BC may be an important pollutant contributing a positive radiative forcing; however, the total magnitude of this forcing is uncertain. Overall, BC is likely not the biggest total forcer in the region. Only considering aerosol, most of the aerosol optical extinction is likely due to non-BC aerosol. Quantifying the contribution to DRE from the various aerosol chemical species in a model with size-resolved aerosol microphysics is challenging and beyond the scope of this study, as removing BC from the model would lead to non-linear effects in the optics calculation (for example, in the core-shell assumption removing BC would alter the total size of particles). However, it is likely that most of the aerosol absorption is due to BC. Assuming an external population, we can estimate that roughly 1-5% of

the total aerosol optical extinction in the Arctic is due to BC. We have added the following statement to Section 3.2:
In the model, these BC mass fractions translate to a contribution of roughly 1-5% of optical extinction from BC to the total aerosol optical extinction in the Arctic (calculated assuming fully externally mixed BC).

*The Conclusion states that this method should be applied globally, but I'm not yet convinced that this method provides any valuable insight. How do your model results compare to satellite remote sensing measurements? That would give an indication of how your new modeling method matches the real world.*

While we do not claim to perfectly constrain the DRE, we do feel we make valuable progress in reporting measurements of BC mixing state and integrating those into the model to reduce the range of estimates of DRE. Comparing to optical measurements (satellite or perhaps AERONET), is an interesting idea but beyond the scope of the present manuscript. As we do not include any direct comparisons that show our method improves model estimates of the DRE, we have removed the sentence from the Conclusions recommending this method be applied globally.

*Technical corrections*

*Page 6 line 15 "and, and"*

Thank you, this has been fixed.

*Page 7 line 2 "can results"*

Thank you, fixed.

*Page 11 line 6 should be "Figure 4b"*

Fixed.

*Page 13 line 32 "0.11" needs units*

Thank you, this has been fixed.

---

## Author Comment (AC4) · 21 Jun 2018

We thank Cenlin He for suggesting additional references and suggesting areas requiring further discussion. The suggested references are quite relevant and helpful to the paper. We have responded to each of the points below and have copied our added text (in underlined font).

*The authors used the SP2 and UHSAS measurements of BC mixing states to provide important observational constraints for model estimates of DRE over the Arctic. This work provides valuable information to advance the current understanding. I have a few comments/suggestions.*

*1. Previous observations (China et al., 2015) have shown a large variety of non-core-shell BC coating/mixing structures/morphology. Recent modeling studies (e.g., He et al., 2015, 2016) further indicated a substantial variation (up to a factor of 2) in BC optical properties by using those different coating/mixing structures/morphology. The authors used a core-shell assumption for internally mixed BC, which may lead to some uncertainty. I suggest including these recent studies and add some discussions on this coating/mixing structure issue.*

*References:*

*China, S., et al.: Morphology and mixing state of aged soot particles at a remote marine free troposphere site: Implications for optical properties, Geophys. Res. Lett., 42, 1243–1250, doi:10.1002/2014gl062404, 2015.*

*He, C., et al.: Variation of the radiative properties during black carbon aging: theoretical and experimental intercomparison, Atmos. Chem. Phys., 15, 11967-11980, doi:10.5194/acp-15-11967-2015, 2015.*

*He, C., et al.: Intercomparison of the GOS approach, superposition T-matrix method, and laboratory measurements for black carbon optical properties during aging, J. Quant. Spectrosc. Radiat. Transf., 184, 287–296, doi:10.1016/j.jqsrt.2016.08.004, 2016.*

Thank you for suggesting these citations. We agree it is important to be quite clear on our assumptions regarding BC morphology and have added several statements discussing this assumption. In this study, we seek to constrain only the population mixing state and leave particle morphology to future work. In the introduction of the paper, we attempt to distinguish two types of mixing state: chemical mixing state and morphological mixing state. The former we constrain through observations and the latter we retain the core-shell assumption. We have added the following sentences to our introductory paragraph on morphology to discuss the observed variability in structure and impacts on absorption:

At a remote observation site, China et al. (2015) found substantial variability in the fractal dimension and structure of mixed (i.e., fully encapsulated versus partly encapsulated) BC

particles. Despite this variability, a common assumption for the morphological mixing state of BC is…

However, the degree of absorption enhancement is a strong function of the structure and geometry of the mixed particle as well as the core diameter and shell thickness (He et al., 2015; He et al., 2016).

We have added the following statements to state our measurements only constrain population mixing state:
We use these measurements as constraints on the population mixing state to estimate the direct radiative effect (DRE) in the springtime Arctic, and compare these estimates to the DRE calculated using bounding cases of completely external or internal mixing state-assumptions. Note, these measurements do not allow us to constrain the morphological mixing state.

In Section 2.5.3, where we discuss mixing states, we have added the following sentence explicitly stating that we constrain only the population mixing state:
These measurements constrain only the population mixing state. For cases of mixed BC, we assume an ideal core-shell mixture, but note morphological mixing state is an important uncertainty (China et al., 2015).

In Section 3.4, where we discuss study limitations, we state morphology as a limiting assumption. We have added the citation to China et al. (2015) here as well:
First, the measurements described here constrain only population mixing state. With regard to particle morphology, we assume a core-shell configuration with BC at the exact center of the particle. Several studies have suggested this may not always be representative of atmospheric aerosol (e.g., Cappa et al., 2012; China et al., 2015).

*2. Page 3, Lines 13-15: For the authors' information, a recent study (Qi et al., 2017a) also used the GEOS-Chem adjoint model to investigate the BC source contributions to the Arctic, which included most recent BC emissions (e.g., gas flaring, small-fire biomass burning, anthropogenic). This recent study is a useful reference that can be cited here.*

*References:*

*Qi, L., et al.: Sources of Springtime Surface Black Carbon in the Arctic: An Adjoint Analysis for April 2008, Atmos. Chem. Phys., 17, 9697-9716, doi:10.5194/acp-17- 9697-2017, 2017a.*

Yes, thank you, this is a very helpful citation for this paragraph. We have added a citation to Qi et al. (2017) to the list of citations in Line 16. We have also added the following sentence to the same paragraph:

Similarly, Qi et al. (2017) found BC concentrations in the Arctic in April 2008 to be largely from anthropogenic sources in Asia and biomass burning sources in Siberia.

*3. Section 2.5.1: recent studies (Qi et al., 2017b,c) have improved BC dry and wet deposition schemes in GEOS-Chem model, which leads to better BC simulations over the Arctic. The authors did not describe the BC dry and wet deposition scheme used in the present study, which are key factors influencing BC simulations. I suggest adding some descriptions on BC deposition scheme used in the model and also including some discussions associated with the aforementioned recent improvements.*

*References:*

*Qi, L., et al.: Factors controlling black carbon distribution in the Arctic, Atmos. Chem. Phys., 17, 1037-1059, doi:10.5194/acp-17-1037-2017, 2017b.*

*Qi, L., et al.: Effects of the Wegener-Bergeron-Findeisen Process on Global Black Carbon Distribution, Atmos. Chem. Phys., 17, 7459-7479, doi:10.5194/acp-17-7459- 2017, 2017c.*

We suspect that the dry and wet deposition schemes are likely the cause of the lower BC mass concentrations in GEOS-Chem-TOMAS relative the observations. The deposition scheme for GEOS-Chem-TOMAS is different from that in the standard GEOS-Chem model. As a measurement-model comparison is not the focus of this paper, we leave detailed descriptions of the aerosol microphysical routines in TOMAS to other publications (and we cite them in the relevant section). We have, however, added a sentence to Section 3.2 discussing Qi et al. (2017b,c) and the importance of wet and dry deposition:
The lower BC mass concentrations may be due to the representation of dry and wet scavenging in TOMAS, which does not include recent updates described in Qi et al. (2017b,c).

---

## Author Response (AR1)

[revised manuscript text omitted]

**TOMAS output (250 nm size bin):**
**0.06 µg m⁻³ of BC mass;**
**1.4 µg m⁻³ total aerosol mass**
**100 particles cm⁻³**

*a).* external and external*1.5

*b).* $r_{shell}$-constrained

*c).* fBC-constrained

*d).* allCoreShell

| | | | |
|---|---|---|---|
| BC core diameter: 250 nm | BC core diameter: 150 nm | BC core diameter: 128 nm | BC core diameter: 86 nm |
| **Shell thickness:  0 nm** | **Shell thickness:  50 nm** | Shell thickness:  61 nm | Shell thickness:  82 nm |
| BC particles:  4% | BC particles:  20% | **BC particles:  30%** | **BC particles:  100%** |

**Figure 2. Schematic presenting the different BC mixing states for the TOMAS size bin corresponding to particle diameters of 250 nm. The bold text shows the parameter being constrained in each mixing state. In this example, GEOS-Chem-TOMAS simulates 4% BC mass fraction and a particle number concentration of 100 cm⁻³. In the *external* mixing-state assumption, all BC mass forms separate particles (rounded to 1 particle out of 10 for convenience), while in the *allCoreShell* mixing-state assumption, all BC mass is spread among all particles. The $r_{shell}$-*constrained* mixing state uses SP2 measurements to constrain BC core diameter and shell thickness. The *fBC-constrained* mixing state uses BC-containing particle fractions from the SP2 and UHAS as the constraint on mixing state.**

Jack 5/23/2018 2:31 PM

Jack 5/23/2018 2:31 PM

[Figure]

**Figure 3. Coating thickness as a function of rBC core diameter. Grey markers are the median, dark shaded area is the 25th-75 percentile, lighter shaded area is the 10-90th percentile of coating thicknesses for each bin.**

[Figure]

**Figure 4. Number distributions for the bare rBC cores (black), the coated rBC-containing particles (green), and the total aerosol as detected by the UHSAS (grey) for the median coating thickness scenario. Shown in the bottom panel is the distribution of the fraction of the total aerosol containing rBC in the median coating scenario. The solid black line is a polynomial fit to the observations following the form (with x in nm): $y = 1.25 - (1.52 \times 10^{-2}) \times x + (8.05 \times 10^{-5}) \times x^2 - (2.00 \times 10^{-7}) \times x^3 + (2.31 \times 10^{-10}) \times x^4 - (1.00 \times 10^{-13}) \times x^5$.**

Jack 6/18/2018 2:03 PM

Jack 6/18/2018 2:03 PM

Jack 6/18/2018 2:03 PM

Jack 6/18/2018 2:04 PM

Jack 6/18/2018 2:04 PM

Jack 6/18/2018 2:04 PM

Jack 6/18/2018 2:05 PM

Jack 6/18/2018 2:05 PM

Jack 6/18/2018 2:05 PM

[Figure]

**Figure 5.** The fraction of BC aerosol mass relative to total aerosol mass at each size bin in GEOS-Chem-TOMAS at three vertical levels along with the fraction determined from the SP2 and UHSAS observations.

Jack 5/25/2018 12:30 PM

[Figure]

[Figure]

[Figure]

Jack 5/25/2018 12:33 PM

**Figure 6. The simulated pan-Arctic mean shell thickness as a function of BC core diameter when the number fraction of BC is constrained by observations (left), and the simulated number fraction of BC when shell thickness as a function of core diameter is constrained by observations (right).**

[Figure]

**Figure 7. The net direct radiative effect over the Arctic in April assuming externally mixed particles.**

[Figure]

**Figure 8.** The difference in DRE between *fBC-constrained* (a,b,c) and *$r_{shell}$-constrained* (d,e,f) compared to the bounding mixing-state assumptions.